# FSI-Edit: Frequency and Stochasticity Injection for Flexible Diffusion-Based Image Editing

**Kaixiang Yang[†], Xin Li[†], Yuxi Li[†], Qiang Li, Zhiwei Wang[*]**

Wuhan National Laboratory for Optoelectronics, Huazhong University of Science and Technology

[†]: Co-first authors, [*]: Corresponding author.

{kxyang, lixin2023, liyuxi9, liqiang8, zwwang}@hust.edu.cn

**Non-Rigid Editing**        **Rigid Editing**

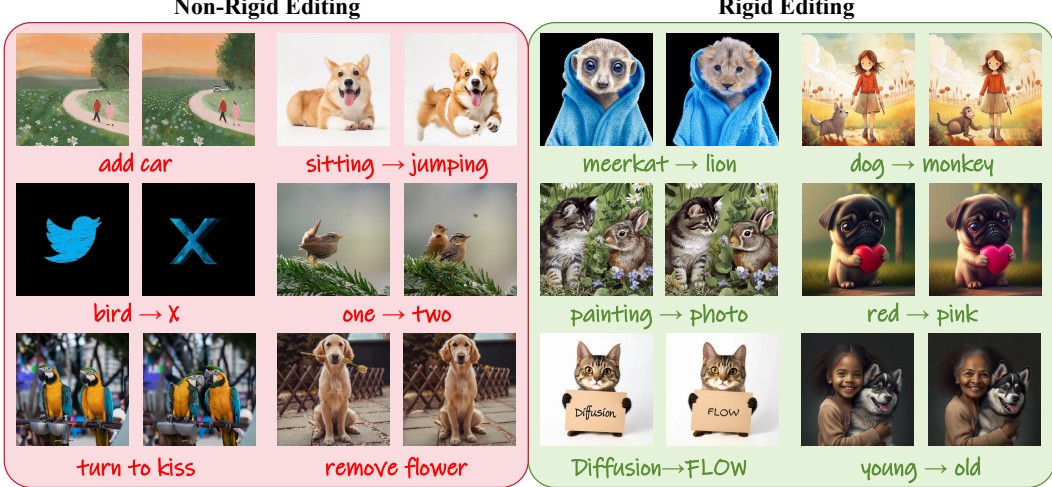

Figure 1: Our method is capable of handling non-rigid editing, including pose changes, object addition, and removal, as well as rigid editing.

## Abstract

Latent Diffusion-based Text-to-Image (T2I) is a free image editing tool that typically reverses an image into noise, reconstructs it using its original text prompt, and then generates an edited version under a new target prompt. To preserve unaltered image content, features from the reconstruction are directly injected to replace selected features in the generation. However, this direct replacement often leads to feature incompatibility, compromising editing fidelity and limiting creative flexibility, particularly for non-rigid edits (*e.g.*, structural or pose changes). In this paper, we aim to address these limitations by proposing **FSI-Edit**, a novel framework using frequency- and stochasticity-based feature injection for flexible image editing. First, FSI-Edit enhances feature consistency by injecting *high-frequency* components of reconstruction features into generation features, mitigating incompatibility while preserving the editing ability for major structures encoded in low-frequency information. Second, it introduces controlled *noise* into the replaced reconstruction features, expanding the generative space to enable diverse non-rigid edits beyond the original image's constraints. Experiments on non-rigid edits, *e.g.*, addition, deletion, and pose manipulation, demonstrate that FSI-Edit outperforms existing baselines in target alignment, semantic fidelity and visual quality. Our work highlights the critical roles of frequency-aware design and stochasticity in overcoming rigidity in diffusion-based editing.

39th Conference on Neural Information Processing Systems (NeurIPS 2025).

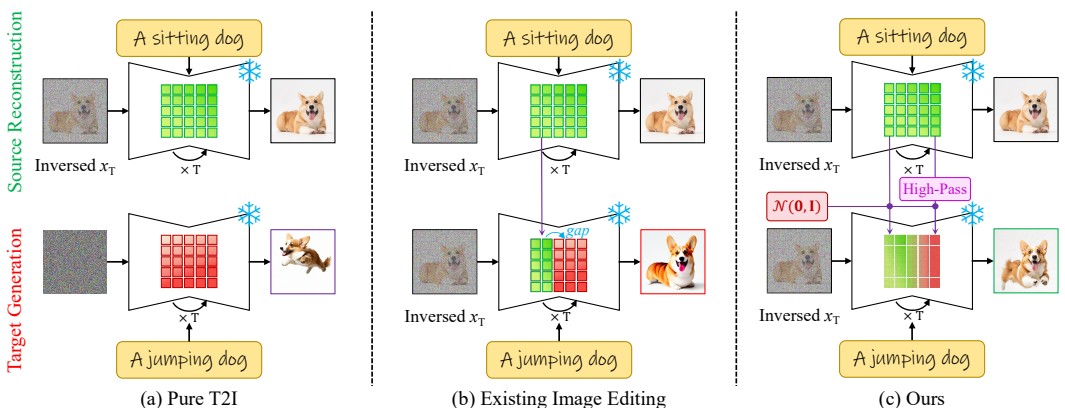

Figure 2: Editing Paradigm Comparison. Top: source reconstruction; Bottom: target generation. (a) Pure T2I: there is no interaction between source and target images, results are random and unrelated to the source. (b) Existing Image Editing: typical editing methods directly inject features, causing semantic gaps and low flexibility, especially for non-rigid edits. (c) Ours: injects only high-frequency residuals and adds stochasticity, reducing semantic gap and improving edit quality.

# 1   Introduction

Diffusion models [1] have achieved remarkable success in the domain of Text-to-Image (T2I) generation in recent advances [2–7]. In particular, Latent Diffusion Models (LDMs) demonstrate exceptional ability to translate textual descriptions (*i.e.,* prompts) into high-quality images, leading to their widespread adoption in downstream applications such as image [8, 9] and video editing [10, 11]. Image editing refers to the task of transforming a source image into a desired target image guided by user-provided prompts. This technology has become an integral part of daily life, with broad applications across domains such as social media and visual effects.

The typical workflow of LDM-based image editing follows a common paradigm. First, the source image is inverted back into a latent noise representation, typically through DDIM inversion [12] or more advanced strategies [13–15]. From this latent noise, two parallel denoising processes are initiated: one reconstructs the original image conditioned on the source prompt, while the other generates a modified version guided by a new target prompt. A key challenge is balancing the preservation of the original content with the incorporation of desired semantic changes, which is typically achieved via information replacement from the reconstruction stream to the generation stream.

A simple way is partially blending the latent noise of the source image with random noise at the beginning of the generation process, often in the frequency domain [16]. Recent methods [8, 9, 17–20] have further explored intermediate feature-level replacement, as shown in Figure 2b. They focus primarily on manipulating query, key, and value features within the self-attention or cross-attention layers, enabling more precise and controllable editing. Specifically, during target generation, selected attention layers are identified, and a subset of their features (*e.g.,* query, key, or value) is directly replaced with corresponding features from the reconstruction stream. This provides visual guidance for preserving image regions that are not intended to be edited.

While existing methods perform well in rigid editing tasks, non-rigid editing such as object addition, removal, or significant pose alteration often requires substantial modifications not only to the target object but also to its surrounding context. In such cases, the editing region naturally extends beyond the object itself, and the direct feature replacement widely-employed in most methods faces two major limitations. **First**, a *semantic gap* exists between the reconstruction and generation processes. Injecting attention features from the reconstruction branch often does not faithfully preserve the original content and can introduce artifacts or structural distortions, especially when editing involves significant pose or viewpoint changes. **Second**, non-rigid edits require fully leveraging the *generative capacity* of the base model. Semantic guidance alone is often insufficient to overcome the strong inductive bias of the original image, which restricts the model's ability to perform flexible and meaningful content transformations. As a result, existing approaches struggle to achieve both content fidelity and effective structural reshaping in non-rigid editing tasks.

To address these issues, we propose **FSI-Edit**, a novel tuning-free framework that enables more flexible and effective image editing through frequency- and stochasticity-based feature injection, as shown in Figure 2c. To bridge the semantic gap between the reconstruction and generation features, we introduce a *frequency residual fusion* mechanism. It allows FSI-Edit to selectively

inject only the high-frequency components from reconstruction features into the generation stream. This preserves fine-grained textures from the original image while avoiding interference from low-frequency structural information, thereby supporting more expressive non-rigid edits. Furthermore, to fully activate the generative capacity of the base diffusion model, we incorporate *stochastic noise injection* during generation. This controlled perturbation enriches the latent space and empowers the model to perform substantial structural changes while maintaining semantic alignment.

We summarize our contributions as follows:

- 1) We identify and address two core limitations of most current image editing methods for non-rigid edits, *e.g.,* semantic inconsistency between reconstruction and generation features, and insufficient generative flexibility due to strong image priors.

- 2) We propose FSI-Edit, featuring a new frequency residual fusion module that selectively transfers high-frequency details for more accurate feature alignment, and a stochastic noise injection strategy that expands the generation space to enable more precise and flexible structural transformations.

- 3) We conduct extensive experiments on PIE-Bench benchmark, and the comparison results demonstrate that FSI-Edit significantly outperforms existing methods on both rigid and non-rigid editing tasks, confirming its effectiveness and generalizability across diverse editing scenarios.

## 2 Related Works

### 2.1 Image Editing with Feature Replacement

The field of text-to-image (T2I) generation has advanced rapidly in recent years, driven by powerful models such as DALL-E [21, 3, 22], Imagen [4], Stable Diffusion [5], and Diffusion Transformer (DiT) [7]. These models have enabled a wide range of applications, among which image editing has emerged as a prominent research direction. Early editing approaches [17, 23–26] typically rely on fine-tuning the generative model on the source image using editing instructions. However, such methods often suffer from limited generalization and require additional training overhead.

To improve flexibility and generalization, a line of tuning-free methods has been proposed. These approaches commonly exploit feature interaction between the source image reconstruction and the target image generation. For example, Prompt-to-Prompt (P2P) [8] introduces cross-attention feature replacement to modify image content according to textual prompts while preserving background fidelity. Plug-and-Play (PnP) [9] further explores the effects of feature replacement at various layers, including residual and attention blocks. While effective for rigid edits, these methods struggle with non-rigid transformations such as pose or structure changes. MasaCtrl [19] improves on this by using attention-based masks to localize and control editing regions, enabling more flexible manipulation within the self-attention mechanism. In parallel, the Transformer-based DiT architecture [7, 27] has opened new opportunities for editing [28, 29]. Methods like FT-Edit [30], RF-Edit [31], and Fireflow [32] focus on improving inversion and editing precision. DCEdit [33] utilizes attention-based localization to guide prompt-consistent modifications.

Despite their progress, most of these approaches rely on direct feature replacement from the source branch, which can introduce semantic mismatches, artifacts, or structural distortions, especially in non-rigid edits involving significant content change. Furthermore, the reused features often carry a strong prior from the source image, limiting the model's generative flexibility and editing expressiveness. To overcome these limitations, we draw inspiration from frequency-domain analysis. Rather than injecting full features, we selectively transfer only high-frequency residuals from the source, preserving fine-grained textures while minimizing semantic conflicts. Additionally, we introduce stochastic noise injection during generation to expand the latent space, improving the model's adaptability to diverse and complex edits.

### 2.2 Image Editing with Latent Frequency Processing

Recent works have begun exploring frequency-domain operations within the latent space of diffusion models to facilitate more controllable image editing. FlexiEdit [16] suppresses high-frequency components in the DDIM latent corresponding to editable regions, employs a re-inversion mechanism,

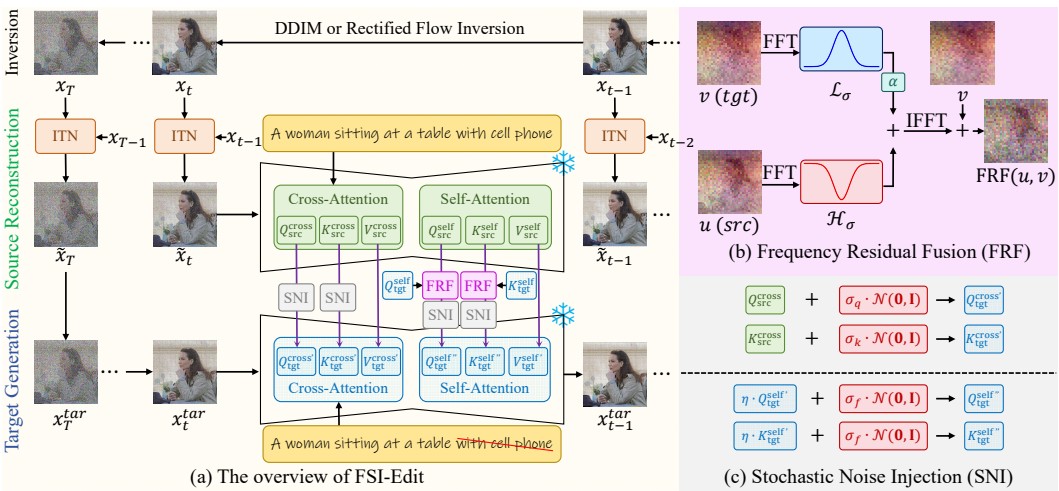

(a) The overview of FSI-Edit

(b) Frequency Residual Fusion (FRF)

(c) Stochastic Noise Injection (SNI)

Figure 3: Overview of FSI-Edit. (a) The full pipeline of FSI-Edit, which consists of three core modules applied in sequence: Frequency Residual Fusion (FRF), Stochastic Noise Injection (SNI), and Inversion Trajectory Navigation (ITN). (b) The structure of FRF, which injects high-frequency source features with low-frequency target features within the self-attention blocks. (c) The SNI module, which injects controlled noise to enhance generative diversity and support non-rigid edits.

and combines attention feature replacement to support non-rigid edits. FreeDiff [34] performs staged frequency-domain filtering during classifier-free guidance (CFG) [35], focusing on attenuating high-frequency noise. However, it requires extensive manual tuning of parameters for each editing scenario. FDS [36] utilizes wavelet decomposition to adaptively select frequency bands based on the editing type, while Gao *et al.* [37] propose a frequency-aware ControlNet module to improve editing controllability.

While these approaches demonstrate the potential of frequency-based manipulation, they operate exclusively in the latent feature space, limiting their ability to address semantic discrepancies between the source and target features. Additionally, their reliance on per-edit parameter tuning hampers practicality and scalability. In contrast, we introduce frequency-domain processing directly at the feature level for the first time. Our method effectively bridges the semantic gap between reconstruction and generation branches without requiring model fine-tuning or laborious hyperparameter adjustment. It generalizes well across both LDM and DiT backbones, offering a robust and efficient solution for high-quality, flexible image editing.

## 3 Method

As shown in Figure 3, FSI-Edit comprises three core components: (1) Frequency Residual Fusion (FRF), (2) Stochastic Noise Injection (SNI), and (3) Inversion Trajectory Navigation (ITN). It builds upon DDIM inversion, rectified flow, and classifier-free guidance (CFG), which are briefly summarized in the Appendix along with architectural details of both FSI-Edit-LDM and FSI-Edit-DiT.

FRF and SNI are applied during the early denoising steps. FRF operates in *self-attention* layers by selectively fusing high-frequency components from the attention triplet features, *i.e.,* query, key, and value, of the source and target branches. This enhances detail preservation while mitigating semantic inconsistencies. SNI is applied to both *self-attention* and *cross-attention* layers, injecting controlled stochasticity into the attention features to promote diversity and enable more flexible, expressive edits. We denote attention features as $\{Q, K, V\}$. Superscripts `self` and `cross` indicate features from self-attention and cross-attention layers, respectively, while subscripts `src` and `tgt` specify whether the features originate from the source reconstruction or target generation stream. Timestep $t$ and layer $l$ indices are omitted for clarity. Implementation details are provided in the Appendix.

### 3.1 Frequency Residual Fusion for Bridging Feature Discrepancy

A central challenge in image editing is reconciling the semantic and structural gap between the source and target representations. This gap becomes especially pronounced in non-rigid edits, where the target must adhere to the prompt while preserving visual consistency with the source.

To address this, we propose Frequency Residual Fusion (FRF), a feature-level harmonization strategy that blends source and target attention features in the frequency domain. Our key idea is to preserve the target's low-frequency components, which encode global layout and structure, while injecting only the high-frequency residuals from the source to retain fine-grained texture and identity.

Let $u \in \mathbb{R}^{C \times H \times W}$ denote features from the source branch and $v \in \mathbb{R}^{C \times H \times W}$ from the target. We construct a frequency-domain fusion $\mathcal{F}_{\text{fuse}}$ via:

$$\mathcal{F}_{\text{fuse}} = \mathcal{H}_\sigma \cdot \text{FFT}(u) + \alpha \cdot \mathcal{L}_\sigma \cdot \text{FFT}(v), \tag{1}$$

where $\text{FFT}(\cdot)$ denotes 2D Fast Fourier Transform, $\mathcal{H}_\sigma, \mathcal{L}_\sigma$ are Gaussian high-pass and low-pass filters with scaling coefficient $\sigma = 0.3$, and fusion weight $\alpha$ is set to 0.2. The fused feature is then mapped back to the spatial domain using inverse FFT $\text{IFFT}(\cdot)$, and combined residually:

$$\text{FRF}(u, v) = \text{IFFT}(\mathcal{F}_{\text{fuse}}) + v. \tag{2}$$

FRF is applied to the Query and Key projections in the self-attention layers to harmonize attention computation:

$$Q_{\text{tgt}}^{\text{self}'} = \text{FRF}(Q_{\text{src}}^{\text{self}}, Q_{\text{tgt}}^{\text{self}}), \quad K_{\text{tgt}}^{\text{self}'} = \text{FRF}(K_{\text{src}}^{\text{self}}, K_{\text{tgt}}^{\text{self}}), \quad V_{\text{tgt}}^{\text{self}'} = V_{\text{src}}^{\text{self}}. \tag{3}$$

This selective frequency blending helps narrow the source–target feature gap, enabling faithful yet flexible edits across diverse content while avoiding semantic drift or prompt misalignment.

## 3.2 Stochastic Noise Injection for Generative Flexibility

Non-rigid editing, especially when involving large structural shifts (*e.g.*, transforming a bird into an "X" shape), requires sufficient generative flexibility. However, existing methods often underutilize this flexibility, resulting in limited editability.

Inspired by Stochastic DDIM [1, 12], which shows that added noise improves inversion diversity, we introduce controlled stochasticity into attention features to enhance expressiveness and support structural transformation. Specifically, we inject Gaussian noise into the Query and Key of the cross-attention layers, while preserving semantic consistency via source-based Value injection:

$$Q_{\text{tgt}}^{\text{cross}'} = Q_{\text{src}}^{\text{cross}} + \sigma_q \cdot \mathcal{N}(\mathbf{0}, \mathbf{I}), \quad K_{\text{tgt}}^{\text{cross}'} = K_{\text{src}}^{\text{cross}} + \sigma_k \cdot \mathcal{N}(\mathbf{0}, \mathbf{I}), \quad V_{\text{tgt}}^{\text{cross}'} = V_{\text{src}}^{\text{cross}}. \tag{4}$$

We further inject noise into frequency-fused Query and Key in self-attention:

$$Q_{\text{tgt}}^{\text{self}''} = \eta \cdot Q_{\text{tgt}}^{\text{self}'} + \sigma_f \cdot \mathcal{N}(\mathbf{0}, \mathbf{I}), \quad K_{\text{tgt}}^{\text{self}''} = \eta \cdot K_{\text{tgt}}^{\text{self}'} + \sigma_f \cdot \mathcal{N}(\mathbf{0}, \mathbf{I}), \tag{5}$$

where $\eta = 0.2$, $\sigma_q = \sigma_k = 0.1$, and $\sigma_f = 0.8$ by default.

Since attention scores depend on relative dot-product magnitudes such as $QK^\top$, injecting bounded noise into frequency-fused features does not affect numerical stability.

This stochasticity allows the model to escape source constraints and adapt to diverse, complex edit patterns. For rigid edits, we anneal $\sigma_q, \sigma_k, \sigma_f$ to 0 and set $\eta \to 1$ to revert to deterministic attention for precise structure preservation.

## 3.3 Inversion Trajectory Navigation for Source Trajectory Refinement

Prior works often directly replace latent features during inversion, which can disrupt structural integrity and cause inconsistencies in the reconstructed source. In Sections 3.1 and 3.2, we introduced frequency residual fusion and stochastic noise injection at the feature level to reduce semantic gaps and enrich generative diversity. Here, we extend these ideas to the latent space by applying a similar fusion strategy during inversion.

During source image inversion (via DDIM [12] or Rectified Flow [28, 29, 38]), we obtain a sequence of latent states $\{x_T, x_{T-1}, \ldots, x_1\}$. Then, we construct a refined sequence $\{\tilde{x}_T, \tilde{x}_{T-1}, \ldots, \tilde{x}_1\}$ for source image reconstruction, where each latent $\tilde{x}_t$ is obtained by blending the high-frequency components of $x_t$ with the low-frequency components of $x_{t-1}$:

$$\tilde{x}_t = \text{IFFT}\big(\mathcal{H}_\sigma \cdot \text{FFT}(x_t) + \mathcal{L}_\sigma \cdot \text{FFT}(x_{t-1})\big) + \sigma_x \cdot \mathcal{N}(\mathbf{0}, \mathbf{I}), \tag{6}$$

where $\sigma = 0.3$, and $\sigma_x = 1e{-}3$ controls the level of injected noise.

The resulting $\{\tilde{x}_t\}$ is used solely to extract source-side features for cross-branch interactions. We initialize both source and target branches with $\tilde{x}_T$. This operation stabilizes the global structure of the source while retaining fine-grained details and introducing slight stochasticity for flexibility.

# 4 Experiment

## 4.1 Experiment Design

**Dataset and Baselines.** To comprehensively evaluate our method on both rigid and non-rigid editing tasks, we conduct experiments on the PIE-Bench [15] benchmark, which contains 700 image-prompt pairs across 10 diverse editing categories. For assessing non-rigid editing performance specifically, such as object addition, deletion, and pose modification, we additionally curate a subset of 300 samples from PIE-Bench that emphasize such editing. Our comparisons include LDM-based methods (*P2P* [8], *PnP* [9], *MasaCtrl* [19], *FlexiEdit* [16], *FreeDiff* [34]) and DiT-based approaches (*RF-Inv* [39], *StableFlow* [40], *RF-Edit* [31], *DCEdit* [33]). All models are tested using their publicly available implementations and default configurations for fair comparison.

**Metrics.** To holistically assess editing performance and background preservation of different methods, we employ six complementary metrics. Structure Distance [41] measure the structural similarity between edited images and original images, while PSNR, LPIPS [42], MSE and SSIM [43] collectively evaluate content preservation in unedited regions. For text-image consistency, we compute CLIP similarity [44] over both the entire image and the edited region. The dataset-provided masks are used to identify the edited regions, but only during evaluation.

**Implementation Details.** All experiments for FSI-Edit-LDM were conducted using Latent Diffusion Model (LDM) [5] v1.5, while FSI-Edit-DiT was built upon DiT [7] v3.5-Medium. We use 50 DDIM steps for inversion, with a CFG scale of 1. During generation, the target branch uses a CFG scale of 7.5. The same settings are applied across both backbones. All experiments were run on a single NVIDIA RTX 4090 GPU with 17 GB memory usage. The full editing pipeline, including inversion and generation, takes 20 seconds per image. Our code is available at `https://github.com/kk42yy/FSI-Edit`.

## 4.2 Comparisons on Diverse Editing Types

Experimental results on the PIE-Bench are summarized in Table 1. Visual comparisons are shown in Figure 4 and Figure 5. These results are generated using our DiT-based version of FSI-Edit, additional examples and results for the LDM-based variant can be found in the Appendix. Across all editing categories, our method achieves a better balance between semantic alignment and content preservation, effectively maintaining the appearance of unedited regions while accurately applying edits. In contrast, existing methods often lean toward one end of this trade-off. For example, P2P tends to preserve background well but struggles with semantic modification. Our approach handles both aspects simultaneously, demonstrating stronger generalization, especially in non-rigid tasks. Among the DiT-based baselines, FSI-Edit-DiT achieves competitive performance using only the lightweight DiT v3.5-Medium model, running on a single RTX 4090 GPU with clear visual improvements over other approaches.

Table 1: Quantitative comparisons on PIE-Bench [15]. P2P [8] achieves the best performance in preserving background and structure, ours attains the highest CLIP similarity score while maintaining competitive background preservation.

| Method | Model | Structure | Background Preservation | | | | CLIP Similarity | |
|---|---|---|---|---|---|---|---|---|
| | | $Distance_{\times 10^3}\downarrow$ | $PSNR\uparrow$ | $LPIPS_{\times 10^3}\downarrow$ | $MSE_{\times 10^4}\downarrow$ | $SSIM_{\times 10^2}\uparrow$ | $Whole\uparrow$ | $Edited\uparrow$ |
| P2P [8] | UNet | **11.65** | **27.22** | **54.55** | **32.86** | **84.76** | 25.02 | 22.10 |
| PnP [9] | | 24.29 | 22.46 | 106.06 | 80.45 | 79.68 | 25.41 | 22.62 |
| MasaCtrl [19] | | 24.70 | 22.64 | 87.94 | 81.09 | 81.33 | 24.38 | 21.35 |
| FlexiEdit [16] | | 22.13 | 25.74 | 80.45 | 58.45 | 82.62 | 25.15 | **22.87** |
| FreeDiff [34] | | 18.70 | 24.73 | 89.76 | 55.32 | 81.68 | 25.03 | 22.12 |
| Ours-LDM | | 15.84 | 24.69 | 88.42 | 52.21 | 81.93 | **25.46** | 22.30 |
| RF-Inv [39] | Transformer | 48.76 | 19.51 | 195.85 | 155.74 | 68.95 | 25.11 | 22.50 |
| StableFlow [40] | | 19.24 | 23.04 | **76.94** | 84.85 | **87.22** | 24.30 | 21.28 |
| RF-Edit [31] | | 24.45 | 24.41 | 113.44 | 56.46 | 83.84 | 25.03 | 22.28 |
| DCEdit [33] | | 22.36 | 25.41 | 94.17 | 48.09 | 85.60 | 25.47 | **22.71** |
| Ours-DiT* | | **13.71** | **26.61** | 85.44 | **36.50** | 86.25 | **25.69** | 22.50 |

* Ours-DiT is built on DiT v3.5-Medium and runs fully on a single RTX 4090 GPU. In contrast, RF-Inv, StableFlow, and RF-Edit rely on FLUX [45] or its variants, which require over 40GB of memory.

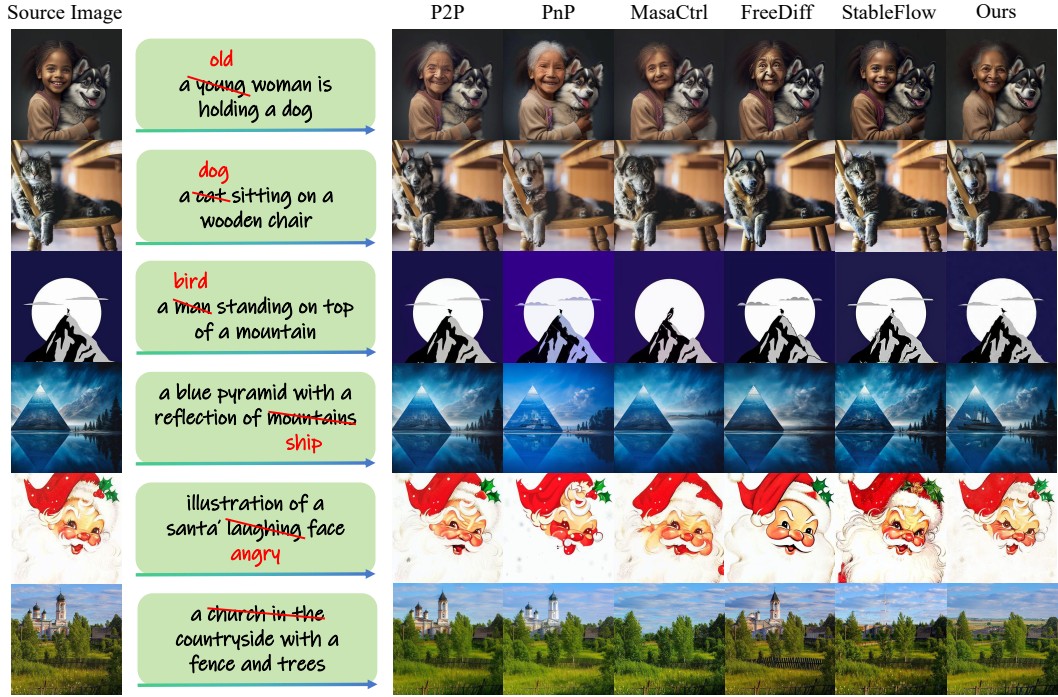

Figure 4: Editing Results on PIE-Bench. Compared to previous methods, our approach better preserves unedited regions, maintaining high consistency with the original image while accurately reflecting the target semantics.

## 4.3 Comparisons on Non-Rigid Editing.

To evaluate non-rigid image editing, which involves structural changes such as addition, deletion, or pose modification, we selected nearly 300 images from the PIE-Bench dataset and constructed corresponding editing prompts. P2P [8] was excluded from the comparison as it is not well-suited for non-rigid editing tasks, whereas DCEdit [33] was excluded, since its official implementation had not been released when our experiments were conducted. Quantitative results are reported in Table 2, while qualitative results are visualized in Figure 5. Due to space constraints, the qualitative results of other methods are provided in Appendix.

Our method consistently delivers high-quality non-rigid edits by accurately aligning with the target prompts while preserving the integrity of unedited regions. For example, in Figure 5 (row 3), our model successfully transforms the bird's shape into an "X" form. In the fifth row, when removing the phone, our method preserves surrounding details such as the vase on the table and the watch on the left wrist. Overall, the effectiveness of our non-rigid editing benefits from the proposed FRF and SNI, which together enhance both semantic alignment and structural fidelity. This allows for precise yet flexible editing while maintaining high realism in unedited areas.

Table 2: Quantitative evaluation of non-rigid editing performance. Non-rigid editing requires substantial modifications to the original image. Our method achieves the highest CLIP similarity score while also maintaining strong background preservation.

| Method | Model | Structure | Background Preservation | | | | CLIP Similarity | |
|---|---|---|---|---|---|---|---|---|
| | | $Distance_{\times 10^3} \downarrow$ | $PSNR \uparrow$ | $LPIPS_{\times 10^3} \downarrow$ | $MSE_{\times 10^4} \downarrow$ | $SSIM_{\times 10^2} \uparrow$ | $Whole \uparrow$ | $Edited \uparrow$ |
| PnP [9] | | 19.23 | 22.17 | 115.86 | 87.83 | 77.47 | 25.10 | 21.32 |
| MasaCtrl [19] | | 22.97 | 22.32 | **94.62** | 87.67 | **79.40** | 24.86 | 20.68 |
| FlexiEdit [16] | UNet | 76.91 | 16.83 | 236.02 | 299.58 | 65.42 | 24.36 | 20.70 |
| FreeDiff [34] | | **18.06** | **23.92** | 101.49 | **64.04** | 78.96 | 25.29 | 21.25 |
| Ours-LDM | | 18.75 | 23.22 | 108.55 | 70.31 | 78.31 | **25.93** | **21.78** |
| RF-Inv [39] | | 54.27 | 18.65 | 253.04 | 188.62 | 62.41 | 26.08 | 22.03 |
| StableFlow [40] | Transformer | **19.58** | 22.40 | **85.63** | 98.60 | **85.19** | 24.93 | 20.84 |
| RF-Edit [31] | | 25.36 | 23.65 | 128.30 | 64.37 | 81.43 | 25.06 | 21.58 |
| Ours-DiT | | 19.87 | **23.91** | 120.82 | **59.95** | 81.71 | **26.58** | **22.16** |

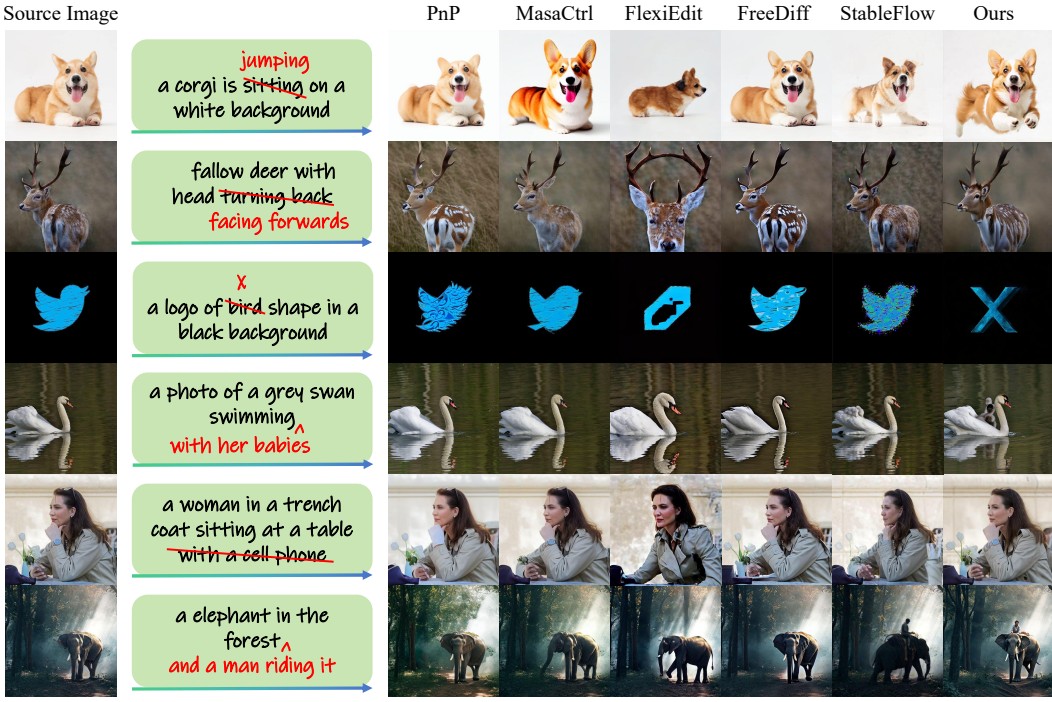

Figure 5: Non-rigid Editing Results. Our method strikes a superior balance between semantic alignment and background preservation, resulting in high-fidelity edits.

## 4.4 User Study

**Setup.** To assess perceptual editing quality, we conducted a user study covering both rigid and non-rigid cases. Eight methods were compared: *PnP, MasaCtrl, FlexiEdit, FreeDiff, RF-Inv, StableFlow, RF-Edit*, and *Ours-DiT*. Participants were instructed to rank results based on (1) faithfulness to the editing instruction and (2) preservation of unedited regions. Each user received around 20 image sets, with method order fully randomized to ensure fairness.

**Results.** We collected **48 valid responses**, including users without prior experience in image editing. Figure 6 reports the ranking distribution, where the *x*-axis represents each method and the *y*-axis shows user ranking (**1 = best**, **8 = worst**). The solid red line denotes the median, while the dashed line indicates the mean.

Methods such as *RF-Inv*, *StableFlow*, and *RF-Edit* show large variance, suggesting unstable performance across different editing types. *PnP*, *MasaCtrl*, *FlexiEdit*, and *FreeDiff* achieve moderate consistency but frequently suffer from semantic drift, as reflected by their higher median.

In contrast, **Ours-DiT achieves the lowest median and mean ranking**, with a compact distribution, indicating strong user agreement. From a perceptual perspective, participants consistently preferred our method for its balance between instruction alignment and source fidelity. These results confirm that FSI-Edit delivers the most reliable and visually convincing edits across diverse scenarios.

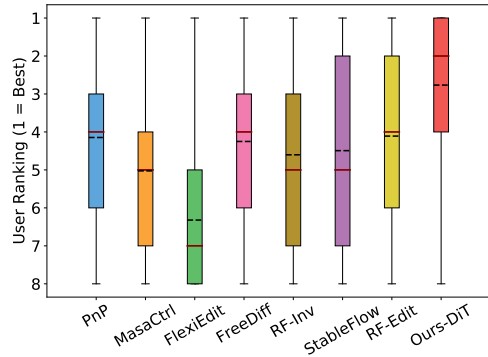

Figure 6: User study ranking distribution across eight editing methods. The *x*-axis lists compared approaches, while the *y*-axis denotes user ranking (1 = best, 8 = worst). The solid red line indicates the median, and the dashed ine represents the mean. Our FSI-Edit achieves the lowest median and most compact variance, showing strong user preference and perceptual robustness.

Table 3: Ablation study on the non-rigid editing dataset. $w/o$ FRF removes the frequency residual fusion module, while $w/o$ SNI denotes the variant without stochastic noise injection, and $w/o$ ITN disables inversion trajectory navigation.

| Method | Structure | Background Preservation | | | | CLIP Similarity | |
|---|---|---|---|---|---|---|---|
| | $Distance_{\times 10^3} \downarrow$ | $PSNR \uparrow$ | $LPIPS_{\times 10^3} \downarrow$ | $MSE_{\times 10^4} \downarrow$ | $SSIM_{\times 10^2} \uparrow$ | $Whole \uparrow$ | $Edited \uparrow$ |
| $w/o$ FRF | 10.51 | 26.97 | 81.33 | 30.88 | 86.38 | 25.50 | 21.14 |
| $w/o$ SNI | 16.38 | 26.12 | 114.22 | 37.13 | 83.52 | 25.85 | 21.27 |
| $w/o$ ITN | 19.90 | 23.92 | 120.14 | 60.47 | 81.72 | 26.47 | 22.05 |
| Ours | 19.87 | 23.91 | 120.82 | 59.95 | 81.71 | 26.58 | 22.16 |

Table 4: Editing performance comparison of ITN with existing reconstruction-refining methods.

| Method | Structure | Background Preservation | | | | CLIP Similarity | |
|---|---|---|---|---|---|---|---|
| | $Distance_{\times 10^3} \downarrow$ | $PSNR \uparrow$ | $LPIPS_{\times 10^3} \downarrow$ | $MSE_{\times 10^4} \downarrow$ | $SSIM_{\times 10^2} \uparrow$ | $Whole \uparrow$ | $Edited \uparrow$ |
| DDIM | 69.43 | 17.87 | 208.80 | 219.88 | 71.56 | 25.01 | 22.44 |
| Direct | 11.65 | 27.22 | 54.55 | 32.86 | 85.10 | 25.02 | 22.10 |
| NT | 13.44 | 27.07 | 59.88 | 35.47 | 84.55 | 24.75 | 21.87 |
| Friend | 11.07 | 26.17 | 58.73 | 38.21 | 84.26 | 25.22 | 22.13 |
| ITN (Ours) | 10.31 | 26.34 | 57.57 | 37.20 | 84.60 | 25.35 | 22.13 |

## 4.5 Ablation Study

**Effects of Key Components.** To assess the contribution of each core component in our method, we conduct ablation studies on the curated non-rigid editing subset of PIE-Bench. Specifically, we evaluate the impact of three modules: Frequency Residual Fusion (FRF), Stochastic Noise Injection (SNI), and Inversion Trajectory Navigation (ITN). We consider three ablated variants of FSI-Edit-DiT: (1) $w/o$ **FRF** removes frequency fusion in self-attention; (2) $w/o$ **SNI** disables stochastic noise injection in attention layers; (3) $w/o$ **ITN** excludes frequency fusion during the source inversion trajectory.

Quantitative results are shown in Table 3, which show that removing either FRF or SNI significantly compromises non-rigid editing quality, leading to weaker prompt alignment and limited structural transformations. In contrast, the full model combines stochasticity to unlock the base model's generative flexibility and FRF to bridge the semantic gap between source and target branches, ultimately achieving a better balance between structural preservation and editing fidelity.

**ITN vs. Reconstruction-refining Methods.** We additionally conduct a dedicated comparison against three widely-used reconstruction-refining methods under the same P2P backbone: Direct Inversion (Direct) [15], Null-text Inversion (NT) [14], and Edit-Friendly Inversion (Friend) [46]. We evaluate both editing fidelity and report the results in Table 4. As shown, ITN achieves the lowest editing distance and the highest whole CLIP alignment, which confirms ITN as an essential component for balancing editability and fidelity in editing tasks.

**Case Study.** Figure 7 illustrates the role of each component through rigid and non-rigid editing examples. Without **FRF**, the model fails to achieve large-scale structural deformation (*e.g.*, turning a bird into an "X" shape). Without **SNI**, new content such as earrings or additional animals is synthesized by reusing existing textures, leading to semantic distortion and local artifacts. Without **ITN**, the model struggles to introduce entirely new objects (*e.g.*, a car or graffiti), resulting in incomplete or collapsed structures. The red circles highlight key regions where the absence of each module causes failure. Together, FRF, SNI, and ITN enable a balance of structural stability, semantic fidelity, and generative diversity, crucial for high-quality image editing.

## 5 Limitations and Broader Impact

### 5.1 Limitation

While our method demonstrates strong performance in non-rigid editing, it may still cause unintended alterations in adjacent regions during fine-grained edits. For instance, in Figure 8a, removing the glasses inadvertently changes the woman's facial expression. Additionally, as shown in Figure 8b, our approach tends to be less responsive to color-specific modifications. In some cases, it may also

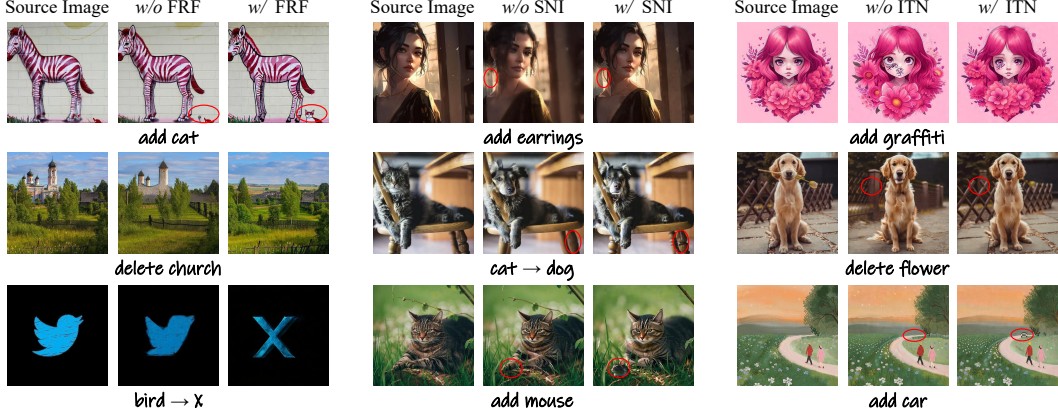

Figure 7: Visual Ablation of Each Module. From left to right, we illustrate the effects of removing FRF, SNI, and ITN, respectively. Without FRF, large-scale non-rigid deformation cannot be achieved, leading to rigid or incomplete edits. Without SNI or ITN, the non-rigid editing quality degrades, and crucial source details may be lost. The red circles highlight the critical differences between using and omitting each module.

fail to fully address all regions implied by the textual prompt, as illustrated in Figure 8c. In future work, we plan to incorporate mask-based spatial guidance, particularly for rigid edits, to enable more localized and accurate modifications. Furthermore, we aim to design frequency fusion strategies tailored to better handle color-sensitive edits, enhancing the controllability of our method in both structural and appearance-level transformations.

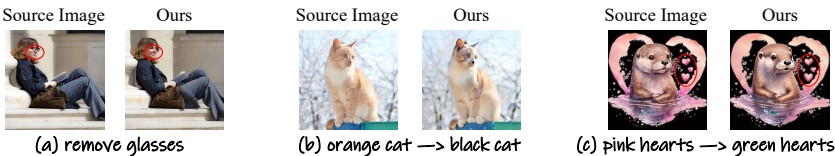

Figure 8: Examples of failed editing cases.

## 5.2 Broader Impact

This work advances the understanding of attention-based feature manipulation in diffusion models by introducing feature-level frequency fusion and stochasticity. These mechanisms enable a more effective balance between background preservation and content modification, resulting in semantically coherent and visually consistent outcomes for both rigid and non-rigid image editing tasks. Looking forward, flexible non-rigid editing with large structural transformations holds potential for practical applications such as visual effects production, where creative structural changes are common, and in surgical imaging, where it can aid in generating physiologically plausible synthetic data for training and simulation. For instance, our method could facilitate the creation of risk-aware surgical scenarios by simulating realistic anatomical deformations. In summary, this research not only improves image editing techniques, but also paves the way for their broader application in creative and scientific domains that demand both precision and flexibility.

## 6 Conclusion

In this paper, we introduced FSI-Edit, a novel tuning-free image editing framework that is effective across both LDM and DiT backbones. FSI-Edit enables flexible and high-fidelity non-rigid editing by incorporating two key mechanisms: (1) frequency residual fusion, which injects high-frequency details from the reconstruction branch to mitigate semantic inconsistency while preserving essential textures, and (2) stochastic noise injection, which expands the generative space and facilitates diverse structural transformations. In addition, FSI-Edit applies frequency-domain fusion not only at the feature level but also along the temporal inversion trajectory, helping preserve unedited content more faithfully while enhancing edit controllability. Extensive experiments on both non-rigid and rigid editing tasks demonstrate that FSI-Edit achieves superior performance, validating its effectiveness, generality, and practical value in a wide range of editing scenarios.

## Acknowledgments and Disclosure of Funding

This work was supported in part by National Natural Science Foundation of China (Grant No.62202189), National Key R&D Program of China (Grant No. 2023YFC2414900), and research grants from Wuhan United Imaging Healthcare Surgical Technology Co., Ltd.

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

# A Preliminary

In this section, we provide a brief overview of the three key preliminaries underlying our tuning-free FSI-Edit framework: DDIM Inversion [12], Rectified Flow [29, 38], and Classifier-Free Guidance [35].

## A.1 DDIM Inversion

DDIM extends DDPM into a non-Markovian diffusion process. In the LDMs, DDIM uses the model's noise estimator $\epsilon_\theta$ to sample the latent $x_{t-1}$ from $x_t$ by:

$$x_{t-1} = \frac{\sqrt{\alpha_{t-1}}}{\sqrt{\alpha_t}}\, x_t - \frac{\sqrt{\alpha_{t-1}(1-\alpha_t)} - \sqrt{(1-\alpha_{t-1})\,\alpha_t}}{\sqrt{\alpha_t}}\, \epsilon_\theta(x_t, t), \tag{7}$$

where $x_t$ denotes the latent noisy features at timestep $t$. By reformulating this discrete update as an ordinary differential equation (ODE), we can apply Euler Integration to solve the reverse process:

$$x_t = \frac{\sqrt{\alpha_t}}{\sqrt{\alpha_{t-1}}}\, x_{t-1} + \frac{\sqrt{\alpha_{t-1}(1-\alpha_t)} - \sqrt{(1-\alpha_{t-1})\,\alpha_t}}{\sqrt{\alpha_{t-1}}}\, \epsilon_\theta(x_{t-1}, t-1). \tag{8}$$

Consequently, when inverting a given source image, we obtain the DDIM Inversion trajectory $\{x_t\}_{t=0}^{T}$.

## A.2 Rectified Flow

[29, 38] learns straight-line transport between two distributions by combining linear interpolation with an ODE-based sampler. Given two observed distributions $x_0 \sim p_0$ and $x_1 \sim p_1$, it defines the continuous trajectory:

$$x_t = t\, x_1 + (1-t)\, x_0, \quad t \in [0,1], \tag{9}$$

and models its time-aware velocity via a neural network $v_\theta(x_t, t)$. In practice, one discretizes time with steps $\{\sigma_t\}$ and applies Euler integration:

$$x_t = x_{t-1} + (\sigma_t - \sigma_{t-1})\, v_\theta(x_{t-1}, t-1). \tag{10}$$

To invert a given endpoint $x_1$ back toward $x_0$, we simply reverse the Euler step:

$$x_{t-1} = x_t + (\sigma_{t-1} - \sigma_t)\, v_\theta(x_t, t-1). \tag{11}$$

By iterating this from $t = 1$ down to $t = 0$, we recover the Rectified Flow inversion trajectory $\{x_t\}_{t=0}^{1}$, which maps an observed source image latent back to its original distribution.

## A.3 Classifier-Free Guidance

To better integrate conditional control into generative diffusion models, Ho *et al.* [35] proposed Classifier-Free Guidance (CFG). CFG replaces the need for an external classifier by interpolating between conditional and unconditional noise predictions. Let $c$ denote the conditioning, so the noise prediction under condition $c$ is $\epsilon_\theta(x_t, c)$, while its unconditional counterpart is $\epsilon_\theta(x_t, \varnothing)$. CFG then computes the guided noise estimate as:

$$\tilde{\epsilon}_\theta(x_t, c) = \epsilon_\theta(x_t, \varnothing) + \lambda\big(\epsilon_\theta(x_t, c) - \epsilon_\theta(x_t, \varnothing)\big), \tag{12}$$

where $\lambda > 1$ is the guidance scale.

# B Implementation Details of FSI-Edit-LDM and FSI-Edit-DiT

We implement our method on top of two popular T2I backbones: Latent Diffusion Models (LDM) [5] v1.5 and Diffusion Transformers (DiT) [7] v3.5-Medium, leading to two variants: FSI-Edit-LDM and FSI-Edit-DiT. The core components, Frequency Residual Fusion (FRF), Stochastic Noise Injection (SNI), and Inversion Trajectory Navigation (ITN), remain consistent across both variants. However, their integration is adapted to accommodate the architectural differences between backbones.

## B.1 FSI-Edit-LDM

Shown in Algorithm 1, for the LDM-based version of our method, we follow the feature interaction strategy of PnP [9] and apply our proposed modules, Frequency Residual Fusion (FRF), Stochastic Noise Injection (SNI), and Inversion Trajectory Navigation (ITN), at specific locations within the UNet architecture:

- FRF: embedded in the feature map of the 4th decoder *residual* block. This is active for the first 80% of the denoising process (*i.e.*, the first 40 timesteps).

- SNI: applied to the *self-attention* blocks from the 4th to the 11th decoder layers of the UNet. We inject noise into the attention queries and keys ($Q_{\mathtt{tgt}}^{\mathtt{self}}$, $K_{\mathtt{tgt}}^{\mathtt{self}}$), while directly injecting the values ($V_{\mathtt{tgt}}^{\mathtt{self}}$). This operation is performed during the first 50% of the denoising steps (*i.e.*, the first 25 timesteps).

- ITN: operated over the entire denoising trajectory, performing frequency-based blending between successive latent states to extract source-consistent features for cross-branch editing.

---

**Algorithm 1** FSI-Edit-LDM

---

1: **Input:** origin image $x_0$, inversion steps $T$, denoising model $\epsilon_\theta$, source target prompts $\mathcal{P}_{src}$, $\mathcal{P}_{tgt}$, *res-block* and *self-attention* thresholds $\tau_{res}$ and $\tau_{self}$
2: **Stage I: DDIM Inversion**
3: **for** $t = 1, \cdots, T$ **do**
4: $\quad x_t = \frac{\sqrt{\alpha_t}}{\sqrt{\alpha_{t-1}}} x_{t-1} + \frac{\sqrt{\alpha_{t-1}(1-\alpha_t)} - \sqrt{(1-\alpha_{t-1})\alpha_t}}{\sqrt{\alpha_{t-1}}} \epsilon_\theta(x_{t-1}, t-1, \mathcal{P}_{src})$
5: **end for**
6: Get the inversion trajectory $\{x_t\}_{t=1}^T$
7: **Stage II: FSI Editing**
8: $x_T^{tar} = \mathrm{ITN}(x_T, x_{T-1})$
9: **for** $t = T, \cdots, 1$ **do**
10: $\quad \tilde{x}_t = \mathrm{ITN}(x_t, x_{t-1})$
11: $\quad f_{t,\mathtt{src}}^{\mathtt{res}}, Q_{t,\mathtt{src}}^{\mathtt{self}}, K_{t,\mathtt{src}}^{\mathtt{self}}, V_{t,\mathtt{src}}^{\mathtt{self}} \leftarrow \epsilon_\theta(\tilde{x}_t, t, \mathcal{P}_{tgt})$
12: $\quad$ **if** $t > \tau_{res}$ **then**
13: $\quad\quad f_{t,\mathtt{tgt}}^{\mathtt{res}''} = \mathrm{SNI}\big(\mathrm{FRF}(f_{t,\mathtt{src}}^{\mathtt{res}}, f_{t,\mathtt{tgt}}^{\mathtt{res}})\big)$
14: $\quad$ **else**
15: $\quad\quad f_{t,\mathtt{tgt}}^{\mathtt{res}''} = f_{t,\mathtt{tgt}}^{\mathtt{res}}$
16: $\quad$ **end if**
17: $\quad$ **if** $t > \tau_{self}$ **then**
18: $\quad\quad Q_{t,\mathtt{tgt}}^{\mathtt{self}''}, K_{t,\mathtt{tgt}}^{\mathtt{self}''} = \mathrm{SNI}\big(\mathrm{FRF}(Q_{t,\mathtt{src}}^{\mathtt{self}}, K_{t,\mathtt{src}}^{\mathtt{self}})\big) ; V_{t,\mathtt{tgt}}^{\mathtt{self}'} = V_{t,\mathtt{src}}^{\mathtt{self}}$
19: $\quad$ **else**
20: $\quad\quad Q_{t,\mathtt{tgt}}^{\mathtt{self}''}, K_{t,\mathtt{tgt}}^{\mathtt{self}''}, V_{t,\mathtt{tgt}}^{\mathtt{self}'} = Q_{t,\mathtt{tgt}}^{\mathtt{self}}, K_{t,\mathtt{tgt}}^{\mathtt{self}}, V_{t,\mathtt{tgt}}^{\mathtt{self}}$
21: $\quad$ **end if**
22: $\quad x_{t-1}^{tar'} = \epsilon_\theta(x_t^{tar}, t, \mathcal{P}_{tgt}; f_{t,\mathtt{tgt}}^{\mathtt{res}''}, Q_{t,\mathtt{tgt}}^{\mathtt{self}''}, K_{t,\mathtt{tgt}}^{\mathtt{self}''}, V_{t,\mathtt{tgt}}^{\mathtt{self}'})$
23: $\quad x_{t-1}^{tar} = \mathrm{DDIM\text{-}Samp}(x_t^{tar}, x_{t-1}^{tar'})$
24: **end for**
25: **Output:** Editing image $x_0^{tar}$

---

## B.2 FSI-Edit-DiT

As illustrated in Algorithm 2, we adopt DiT v3.5-Medium (bfloat16) as the backbone for our transformer-based implementation. FSI-Edit-DiT incorporates all three key modules with the following configurations:

- FRF: embedded to the *self-attention* layers of the 0th to 12th Transformer blocks, and is active during the first 50% of the denoising timesteps.

- SNI: applied to all *cross-attention* layers throughout the Transformer blocks, and is active for the first 50% of the denoising steps.

- ITN: performed across all timesteps to refine source latent representations

**Algorithm 2** FSI-Edit-DiT

---

1: **Input:** origin image $x_0$, inversion steps $T$, velocity field $v_\theta$, source target prompts $\mathcal{P}_{src}, \mathcal{P}_{tgt}$, *cross-block* and *self-attention* thresholds $\tau_{cross}$ and $\tau_{self}$
2: **Stage I: Rectified Flow Inversion**
3: **for** $t = 1, \cdots, T$ **do**
4: $\quad x_t = x_{t-1} + (\sigma_t - \sigma_{t-1})v_\theta(x_{t-1}, t-1, \mathcal{P}_{src})$
5: **end for**
6: Get the inversion trajectory $\{x_t\}_{t=1}^{T}$
7: **Stage II: FSI Editing**
8: $x_T^{tar} = \text{ITN}(x_T, x_{T-1})$
9: **for** $t = T, \cdots, 1$ **do**
10: $\quad \tilde{x}_t = \text{ITN}(x_t, x_{t-1})$
11: $\quad (Q_{t,\text{src}}^{\text{cross}}, K_{t,\text{src}}^{\text{cross}}, V_{t,\text{src}}^{\text{cross}}), (Q_{t,\text{src}}^{\text{self}}, K_{t,\text{src}}^{\text{self}}, V_{t,\text{src}}^{\text{self}}) \leftarrow v_\theta(\tilde{x}_t, t, \mathcal{P}_{tgt})$
12: $\quad$ **if** $t > \tau_{cross}$ **then**
13: $\quad\quad Q_{t,\text{tgt}}^{\text{cross}'}, K_{t,\text{tgt}}^{\text{cross}'}, V_{t,\text{tgt}}^{\text{cross}'} = \text{SNI}\big(Q_{t,\text{src}}^{\text{cross}}, K_{t,\text{src}}^{\text{cross}}, V_{t,\text{src}}^{\text{cross}}\big)$
14: $\quad$ **else**
15: $\quad\quad Q_{t,\text{tgt}}^{\text{cross}'}, K_{t,\text{tgt}}^{\text{cross}'}, V_{t,\text{tgt}}^{\text{cross}'} = Q_{t,\text{tgt}}^{\text{cross}}, K_{t,\text{tgt}}^{\text{cross}}, V_{t,\text{tgt}}^{\text{cross}}$
16: $\quad$ **end if**
17: $\quad$ **if** $t > \tau_{self}$ **then**
18: $\quad\quad Q_{t,\text{tgt}}^{\text{self}''}, K_{t,\text{tgt}}^{\text{self}''} = \text{SNI}\big(\text{FRF}(Q_{t,\text{src}}^{\text{self}}, K_{t,\text{src}}^{\text{self}})\big) \, ; V_{t,\text{tgt}}^{\text{self}'} = V_{t,\text{src}}^{\text{self}}$
19: $\quad$ **else**
20: $\quad\quad Q_{t,\text{tgt}}^{\text{self}''}, K_{t,\text{tgt}}^{\text{self}''}, V_{t,\text{tgt}}^{\text{self}'} = Q_{t,\text{tgt}}^{\text{self}}, K_{t,\text{tgt}}^{\text{self}}, V_{t,\text{tgt}}^{\text{self}}$
21: $\quad$ **end if**
22: $\quad x_{t-1}^{tar'} = v_\theta(x_t^{tar}, t, \mathcal{P}_{tgt}; Q_{t,\text{tgt}}^{\text{cross}'}, K_{t,\text{tgt}}^{\text{cross}'}, V_{t,\text{tgt}}^{\text{cross}'}, Q_{t,\text{tgt}}^{\text{self}''}, K_{t,\text{tgt}}^{\text{self}''}, V_{t,\text{tgt}}^{\text{self}'})$
23: $\quad x_{t-1}^{tar} = \text{RectifiedFlow-Samp}(x_t^{tar}, x_{t-1}^{tar'})$
24: **end for**
25: **Output:** Editing image $x_0^{tar}$

---

## C  Comparison Methods and Experimental Setup

In this section, we present the experimental setup and parameter configurations for the baseline methods used in our comparisons.

### C.1  LDM-Based

For P2P [8], PnP [9], and MasaCtrl [19], we adopt DDIM Direct Inversion [15][1] as the inversion backbone. All image editing experiments are conducted using their default parameter settings.

For FlexiEdit [16], we use the official implementation[2]. To support large-scale, consistent batch processing across methods, we follow the configuration most commonly used in the official examples by fixing the reinversion steps to $t_R = 30$. Since FlexiEdit requires a 'blended word' to localize the editing region, we extract this information from corresponding prompts in PIE-Bench [15]. In cases where the model fails to locate the semantic region associated with the 'blended word', we default to editing the entire image. All other parameters remain at their default settings.

For FreeDiff [34], we use the official implementation[3] and adopt the most representative configuration from the official examples, setting the time steps for filter scheduling to $\tau_i = (801, 781, 581)$ and the high-pass filter sizes to $r_t^H = (32, 32, 10, 10)$. All other parameters are kept at their default values.

All LDM-based methods above are implemented using the v1.4 or v1.5 Stable Diffusion backbone and are executed on a single NVIDIA RTX 4090 GPU with 24GB of memory.

---

[1] https://github.com/cure-lab/PnPInversion
[2] https://github.com/kookie12/FlexiEdit
[3] https://github.com/Thermal-Dynamics/FreeDiff

Table 5: User study ranking results (Mean ± Standard Deviation; lower indicates better perceptual quality). Supplement to Fig. 6 in the main paper.

| Method | PnP | MasaCtrl | FlexiEdit | FreeDiff | RF-Inv | StableFlow | RF-Edit | Ours-DiT |
|--------|-----|----------|-----------|----------|--------|------------|---------|----------|
| Ranking | $3.94 \pm 2.22$ | $4.80 \pm 2.30$ | $5.99 \pm 2.59$ | $4.09 \pm 2.22$ | $4.39 \pm 2.42$ | $4.29 \pm 2.51$ | $3.92 \pm 2.29$ | $\mathbf{2.66 \pm 1.96}$ |

## C.2 DiT-Based

We use the official implementation of StableFlow[4] [40], with the inversion steps set to 50. For RF-Inv [39], we adopt the official code[5] with the default settings. For RF-Edit [31], we use the official repository[6], with the guidance scale set to 2 and the injection step set to 5. All other parameters remain at their default values. The above three methods are executed on a single NVIDIA A100-PCIE-80GB GPU.

## D  User Study Statistics

To complement Fig. 6 in the main paper, we report the detailed statistical results of the user study, including the mean and standard deviation of ranking scores for each method. As described in the main text, each participant was asked to rank eight editing methods based on overall quality (semantic alignment and preservation). Lower scores indicate better perceived performance. Table. 5 provides a clearer view of user preference consistency and further confirm the advantage of FSI-Edit-DiT, which shows both the lowest mean rank and the smallest variance.

## E  Extended Ablation Studies

In this section, we investigate how different parameter choices in each module of our method affect editing quality and controllability. We conduct a series of controlled experiments to analyze the sensitivity and contribution of key hyperparameters within the FRF, SNI, and ITN components. Specifically, we set the default values as follows: pairing distance $d = 1$ for ITN, fusion weight $\alpha = 0.2$ and Gaussian scaling coefficient $\sigma = 0.3$ for FRF, noise ratio $\eta = 0.2$ (corresponding to $\sigma_f = 0.8$) for SNI, and FSI-Edit intervention durations of 50% and 65% for non-rigid and rigid editing tasks, respectively.

To ensure fairness, all other parameters are kept fixed at these default values when exploring the effect of any single variable. To directly assess the influence of each parameter on editing behavior, all ablation studies are conducted on the non-rigid editing dataset using the FSI-Edit-DiT.

### E.1  Effect of ITN Pairing Distance

Table 6: Ablation study on the pairing distance $d$ in ITN.

| $d$ | Structure | Background Preservation | | | | CLIP Similarity | |
|-----|-----------|-------------------------|---|---|---|-----------------|---|
| | $Distance_{\times 10^3} \downarrow$ | $PSNR \uparrow$ | $LPIPS_{\times 10^3} \downarrow$ | $MSE_{\times 10^4} \downarrow$ | $SSIM_{\times 10^2} \uparrow$ | $Whole \uparrow$ | $Edited \uparrow$ |
| $w/o$ ITN | 19.90 | 23.92 | 120.14 | 60.47 | 81.72 | 26.47 | 22.05 |
| 1 (Ours) | 19.87 | 23.91 | 120.82 | 59.95 | 81.71 | 26.58 | 22.16 |
| 2 | 19.80 | 23.94 | 119.08 | 60.04 | 81.80 | 26.57 | 22.06 |
| 5 | 19.80 | 23.94 | 119.44 | 60.28 | 81.74 | 26.59 | 22.15 |
| 10 | 19.88 | 23.99 | 119.22 | 59.87 | 81.83 | 26.48 | 22.07 |

In our default ITN design, each latent $x_t$ is fused with its immediate predecessor $x_{t-1}$ to stabilize inversion while preserving detail. Here, we investigate how increasing the pairing distance $d$ affects the quality of source reconstruction and downstream editing.

---

[4]`https://github.com/snap-research/stable-flow`
[5]`https://github.com/LituRout/RF-Inversion`
[6]`https://github.com/wangjiangshan0725/RF-Solver-Edit`

Table 7: Ablation study on the pairing distance $d$ in ITN for **source image reconstruction**.

| $d$ | Structure | Background Preservation | | | | CLIP Similarity |
|---|---|---|---|---|---|---|
| | $Distance_{\times 10^3} \downarrow$ | $PSNR \uparrow$ | $LPIPS_{\times 10^3} \downarrow$ | $MSE_{\times 10^4} \downarrow$ | $SSIM_{\times 10^2} \uparrow$ | $Whole \uparrow$ |
| $w/o$ ITN | 3.06 | 32.30 | 25.35 | 9.06 | 91.40 | 25.97 |
| 1 (Ours) | 3.08 | 32.29 | 25.37 | 9.07 | 91.40 | 25.97 |
| 2 | 3.06 | 32.30 | 25.35 | 9.06 | 91.40 | 25.97 |
| 5 | 3.07 | 32.30 | 25.35 | 9.06 | 91.40 | 25.97 |
| 10 | 3.06 | 32.30 | 25.35 | 9.06 | 91.40 | 25.97 |

Specifically, we vary the reference latent from $x_{t-1}$ to $x_{t-d}$, where $d \in \{1, 2, 5, 10\}$, and apply the same frequency-domain fusion strategy:

$$\tilde{x}_t = \text{IFFT}\left(\mathcal{H}_\sigma \cdot \text{FFT}(x_t) + \mathcal{L}_\sigma \cdot \text{FFT}(x_{t-d})\right) + \sigma_x \cdot \mathcal{N}(0, I). \tag{13}$$

We keep $\sigma = 0.3$, and $\sigma_x = 1e-3$ here.

The results are summarized in Table 6. Additionally, we evaluate the impact of different pairing distances $d$ in ITN on source reconstruction quality, as shown in Table 7. From Table 7, we observe that varying $d$ has negligible effect on source image reconstruction. Combined with Table 6, we find that applying ITN consistently improves the CLIP similarity in the edited regions compared to $w/o$ ITN.

In addition, as a complementary result to Table 4 in the main paper, we further provide a quantitative comparison of source image reconstruction among ITN and other reconstruction-refining methods in Table 8. These results further demonstrate that ITN provides superior reconstruction fidelity, preserving both structural details and semantic consistency.

Table 8: Ablation study on the ITN with other reconstruction-refining methods for **source image reconstruction**.

| $d$ | Structure | Background Preservation | | | | CLIP Similarity |
|---|---|---|---|---|---|---|
| | $Distance_{\times 10^3} \downarrow$ | $PSNR \uparrow$ | $LPIPS_{\times 10^3} \downarrow$ | $MSE_{\times 10^4} \downarrow$ | $SSIM_{\times 10^2} \uparrow$ | $Whole \uparrow$ |
| DDIM | 70.23 | 17.76 | 210.84 | 224.43 | 71.39 | 27.07 |
| Direct | 2.95 | 30.57 | 31.41 | 17.60 | 87.50 | 25.45 |
| NT | 3.30 | 30.17 | 33.50 | 18.94 | 87.13 | 25.50 |
| Friend | 3.12 | 30.37 | 32.66 | 18.21 | 87.19 | 25.55 |
| ITN (Ours) | 2.33 | 30.49 | 31.88 | 17.85 | 87.47 | 25.64 |

## E.2 Ablation Study on Filter Strength and Fusion Weight in FRF

To evaluate the effect of key parameters in Frequency Residual Fusion (FRF), we conduct systematic ablation studies on the Gaussian filter strength. Specifically, the low-pass and high-pass filters $\mathcal{L}_\sigma$ and $\mathcal{H}_\sigma$ are defined as:

$$\mathcal{L}_\sigma = \frac{1}{2\pi\sigma^2}e^{-\frac{r^2}{2\sigma^2}} \in \mathbb{R}^{W \times H}, \quad \mathcal{H}_\sigma = 1 - \mathcal{L}_\sigma \in \mathbb{R}^{W \times H}, \tag{14}$$

where $r$ denotes the frequency distance from the center and $\sigma$ controls degree of Gaussian curve.

We experiment with various values of $\sigma \in \{0.1, 0.3, 0.4, 0.5, 0.6, 0.8, 0.9\}$ to control the frequency selectivity of the filters, and investigate the influence of the fusion weight $\alpha \in \{0.0, 0.1, 0.2, 0.3, 0.5, 0.7, 0.9, 1.0\}$, which modulates the relative contribution of the target's low-frequency components. The frequency-domain fusion is then computed as:

$$\mathcal{F}_{\text{fuse}} = \mathcal{H}_\sigma \cdot \text{FFT}(u) + \alpha \cdot \mathcal{L}_\sigma \cdot \text{FFT}(v), \tag{15}$$

$$\text{FRF}(u, v) = \text{IFFT}(\mathcal{F}_{\text{fuse}}) + v. \tag{16}$$

As shown in Table 9 and Table 10, the value of $\sigma$ and $\alpha$ have little impact on the results. However, setting $\alpha = 0$ leads to a noticeable drop in editing performance, indicating that the lack of low-frequency information from the target feature impairs the quality of non-rigid editing. This also highlights the limitations of directly performing feature injection.

Table 9: Ablation study on the Gaussian filter scaling coefficient $\sigma$ in FRF.

| $\sigma$ | Structure | Background Preservation | | | | CLIP Similarity | |
|---|---|---|---|---|---|---|---|
| | $Distance_{\times 10^3} \downarrow$ | $PSNR \uparrow$ | $LPIPS_{\times 10^3} \downarrow$ | $MSE_{\times 10^4} \downarrow$ | $SSIM_{\times 10^2} \uparrow$ | $Whole \uparrow$ | $Edited \uparrow$ |
| w/o FRF | 10.51 | 26.97 | 81.33 | 30.88 | 86.38 | 25.50 | 21.14 |
| 0.1 | 19.84 | 23.92 | 120.65 | 59.88 | 81.72 | 26.59 | 22.15 |
| 0.3 (Ours) | 19.87 | 23.91 | 120.82 | 59.95 | 81.71 | 26.58 | 22.16 |
| 0.4 | 19.87 | 23.92 | 120.64 | 59.89 | 81.73 | 26.60 | 22.25 |
| 0.5 | 19.73 | 23.93 | 120.19 | 59.74 | 81.76 | 26.60 | 22.13 |
| 0.6 | 19.87 | 23.94 | 119.95 | 59.92 | 81.78 | 26.54 | 22.19 |
| 0.8 | 19.57 | 23.96 | 119.42 | 59.67 | 81.84 | 26.60 | 22.14 |
| 0.9 | 19.62 | 23.97 | 119.10 | 59.59 | 81.59 | 26.59 | 22.18 |

Table 10: Ablation study on the Fusion Weight $\alpha$ in FRF.

| $\alpha$ | Structure | Background Preservation | | | | CLIP Similarity | |
|---|---|---|---|---|---|---|---|
| | $Distance_{\times 10^3} \downarrow$ | $PSNR \uparrow$ | $LPIPS_{\times 10^3} \downarrow$ | $MSE_{\times 10^4} \downarrow$ | $SSIM_{\times 10^2} \uparrow$ | $Whole \uparrow$ | $Edited \uparrow$ |
| w/o FRF | 10.51 | 26.97 | 81.33 | 30.88 | 86.38 | 25.50 | 21.14 |
| 0.0 | 19.08 | 23.98 | 116.44 | 59.27 | 82.03 | 23.39 | 22.04 |
| 0.1 | 19.81 | 23.92 | 120.56 | 59.88 | 81.72 | 26.62 | 22.27 |
| 0.2 (Ours) | 19.87 | 23.91 | 120.82 | 59.95 | 81.71 | 26.58 | 22.16 |
| 0.3 | 19.81 | 23.92 | 120.71 | 59.88 | 81.72 | 26.54 | 22.21 |
| 0.5 | 19.90 | 23.92 | 120.56 | 59.85 | 81.72 | 26.61 | 22.22 |
| 0.7 | 19.92 | 23.92 | 120.74 | 59.94 | 81.72 | 26.57 | 22.17 |
| 0.9 | 19.90 | 23.92 | 120.61 | 59.87 | 81.72 | 26.61 | 22.25 |
| 1.0 | 19.85 | 23.92 | 120.64 | 59.92 | 81.71 | 26.58 | 22.19 |

### E.3 Ablation Study on the Noise Mixing Coefficient in SNI

To investigate the effect of the noise-content trade-off in our Stochastic Noise Injection (SNI) module, we conduct ablation experiments on the mixing coefficient $\eta$ in the query and key injection formulation:

$$Q_{\texttt{tgt}}^{\texttt{self}''} = \eta \cdot Q_{\texttt{tgt}}^{\texttt{self}'} + \sigma_f \cdot \mathcal{N}(\mathbf{0}, \mathbf{I}), \quad K_{\texttt{tgt}}^{\texttt{self}''} = \eta \cdot K_{\texttt{tgt}}^{\texttt{self}'} + \sigma_f \cdot \mathcal{N}(\mathbf{0}, \mathbf{I}). \quad (17)$$

To ensure a balanced contribution between deterministic content and injected noise, we fix $\eta + \sigma_f = 1$ and vary $\eta$ across $\{0.0, 0.1, 0.2, 0.3, 0.5, 0.7, 0.9, 1.0\}$. This allows us to systematically evaluate how different levels of stochasticity affect the flexibility and consistency of non-rigid edits.

Table 11: Ablation study on the Noise Mixing Coefficient $\eta$ in SNI. $w/o$ SNI corresponds to the setting where $\eta = 1.0$, and no Gaussian noise is injected into the query and key of the cross-attention layers in the target branch.

| $\eta$ | Structure | Background Preservation | | | | CLIP Similarity | |
|---|---|---|---|---|---|---|---|
| | $Distance_{\times 10^3} \downarrow$ | $PSNR \uparrow$ | $LPIPS_{\times 10^3} \downarrow$ | $MSE_{\times 10^4} \downarrow$ | $SSIM_{\times 10^2} \uparrow$ | $Whole \uparrow$ | $Edited \uparrow$ |
| w/o SNI | 16.38 | 26.12 | 114.22 | 37.13 | 83.52 | 25.85 | 21.27 |
| 0.0 | 19.27 | 23.92 | 118.06 | 59.71 | 81.90 | 26.64 | 22.12 |
| 0.1 | 19.42 | 23.93 | 118.41 | 59.56 | 81.86 | 26.60 | 22.15 |
| 0.2 (Ours) | 19.87 | 23.91 | 120.82 | 59.95 | 81.71 | 26.58 | 22.16 |
| 0.3 | 20.46 | 23.88 | 123.56 | 60.37 | 81.48 | 26.47 | 22.19 |
| 0.5 | 21.73 | 23.77 | 130.38 | 61.52 | 80.88 | 26.50 | 22.23 |
| 0.7 | 23.04 | 23.71 | 137.19 | 62.57 | 80.37 | 26.50 | 22.14 |
| 0.9 | 24.45 | 23.64 | 144.15 | 63.02 | 80.18 | 26.58 | 22.25 |
| 1.0 | 24.88 | 23.60 | 147.03 | 63.04 | 80.14 | 26.53 | 21.99 |

As shown in Table 11, varying $\eta$ has limited influence on the CLIP similarity of the edited regions, indicating stable editing performance. However, setting $\eta = 1.0$, which corresponds to the absence

of randomness, leads to suboptimal editing quality and background preservation. To better leverage the refined source and target branch features for editing, we ultimately choose $\eta = 0.2$ and $\sigma_f = 0.8$ as our default settings.

### E.4 Ablation Study on the Duration of FRF and SNI Interventions

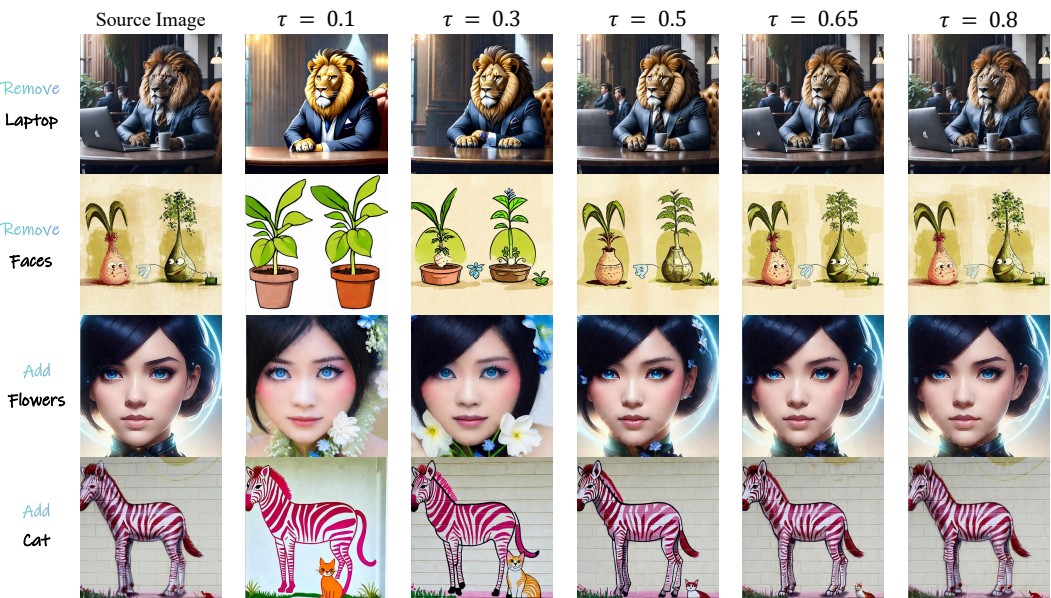

Figure 9: Effect of the duration $\tau$ of the intervention for FRF and SNI. As $\tau$ increases, the target images better preserve source background structures and maintain semantically meaningful edits. However, excessively large $\tau$ may overly constrain generation, limiting the extent of semantic transformations.

To understand the impact of the duration of the intervention for FRF and SNI, we conduct ablation studies by varying the proportion of timesteps $\tau$ during which these modules are applied. Specifically, we experiment with 10%, 30%, 50%, 65%, and 80% of the total denoising steps. Note that $\tau = 0\%$ corresponds to random generation, which tends to strictly adhere to the semantics of the target prompt. As the proportion of guided timesteps increases, background preservation improves, but this often comes at the cost of reduced editability.

Table 12 and Figure 9 present the quantitative and qualitative trade-offs between content preservation and editability. Based on these results, we select a duration of 50% for non-rigid editing, and 65% for rigid editing to achieve a balanced performance.

Table 12: Ablation study on the Duration of FRF and SNI Interventions.

| $\tau$ | Structure | Background Preservation | | | | CLIP Similarity | |
|---|---|---|---|---|---|---|---|
| | $Distance_{\times 10^3} \downarrow$ | $PSNR \uparrow$ | $LPIPS_{\times 10^3} \downarrow$ | $MSE_{\times 10^4} \downarrow$ | $SSIM_{\times 10^2} \uparrow$ | $Whole \uparrow$ | $Edited \uparrow$ |
| 0.1 | 87.01 | 14.96 | 292.32 | 402.72 | 64.17 | 27.63 | 23.40 |
| 0.3 | 41.97 | 19.75 | 187.95 | 144.51 | 74.50 | 27.13 | 23.29 |
| 0.5 (Ours) | 19.87 | 23.91 | 120.82 | 59.95 | 81.71 | 26.58 | 22.16 |
| 0.65 | 11.17 | 26.69 | 82.69 | 32.40 | 85.76 | 25.69 | 21.29 |
| 0.8 | 5.48 | 29.94 | 49.90 | 16.25 | 89.35 | 24.39 | 20.22 |

## F Additional Qualitative Comparisons

### F.1 Supplementary Visual Comparisons

In this section, we present additional comparison results that could not be included in the main paper due to space limitations. As shown in the Figure 10, our DiT-based method demonstrates superior editing performance across both non-rigid and rigid tasks. However, the LDM-based version is more constrained by the limitations of its underlying generative backbone, resulting in suboptimal

outputs in certain cases. For example, in the first row, the model incorrectly alters the dog from the original image; in rows 8 and 9, the results fail to reflect the intended semantic edits, indicating the difficulty LDM has with large-scale non-rigid transformations. In the deletion task of row 11, the model introduces unintended changes, such as adding a glove to the woman's hand. Overall, while our method is partially influenced by the choice of backbone, it achieves consistently strong results when built upon DiT v3.5-Medium.

## F.2    Extended Visual Results and Analysis

Figure 11 and Figure 12 provide additional visual comparisons against all baseline methods on both non-rigid and rigid editing tasks. Our approach, in both DiT-based and LDM-based versions, consistently demonstrates superior editing performance across various scenarios.

In particular, Figure 12 includes failure cases from our main paper (rightmost three columns). As illustrated, most competing methods struggle to preserve fine-grained details and maintain color consistency. In more challenging cases, they fail to execute the intended semantic transformations, such as removing glasses or altering the color of a cat.

## F.3    Diverse Editing Capabilities of FSI-Edit-DiT

To further demonstrate the versatility of our approach, we present qualitative results across a broad spectrum of editing categories. These include rigid edits (*e.g.*, changes in color, material, or style) as well as non-rigid edits (*e.g.*, object addition, removal and pose changes). As shown in Fig. 13 and Fig. 14, our method consistently produces high-fidelity results across all scenarios, underscoring its robustness and generalizability.

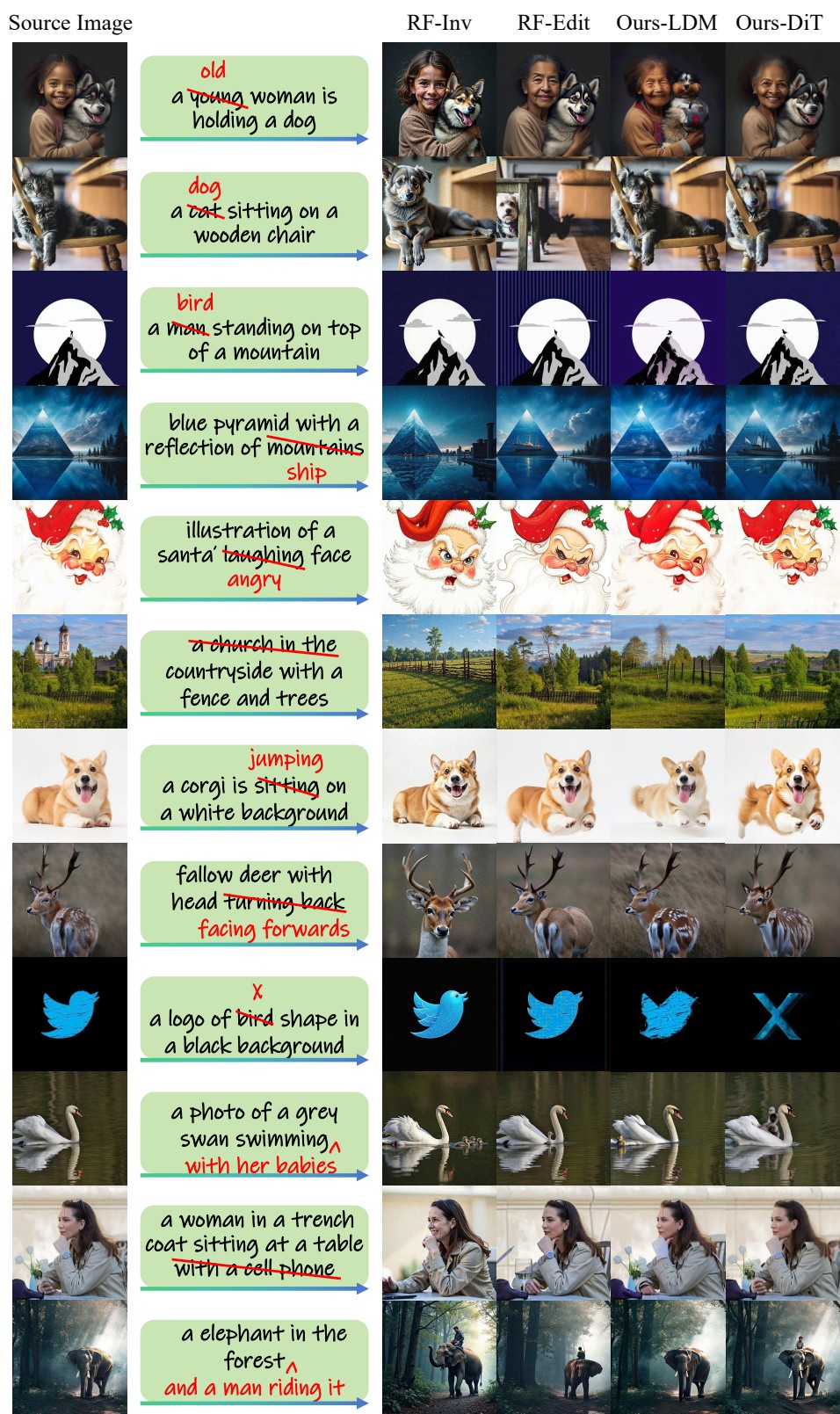

Figure 10: Visual comparison of different methods. This figure supplements the main paper with additional results, including comparisons with RF-Inv, RF-Edit, and our LDM-based approach on both non-rigid and rigid editing tasks.

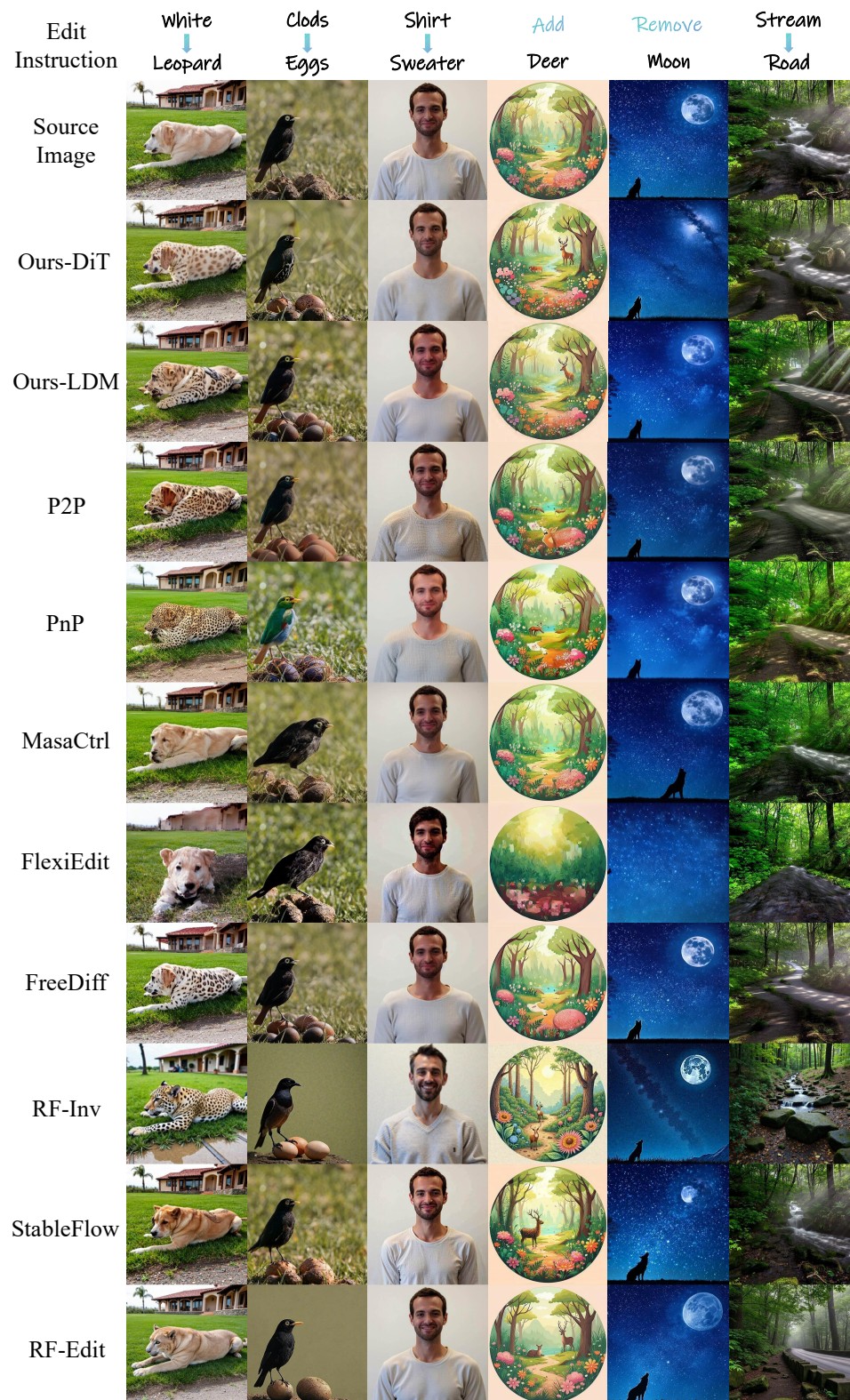

Figure 11: Additional qualitative editing results on PIE-Bench Part I.

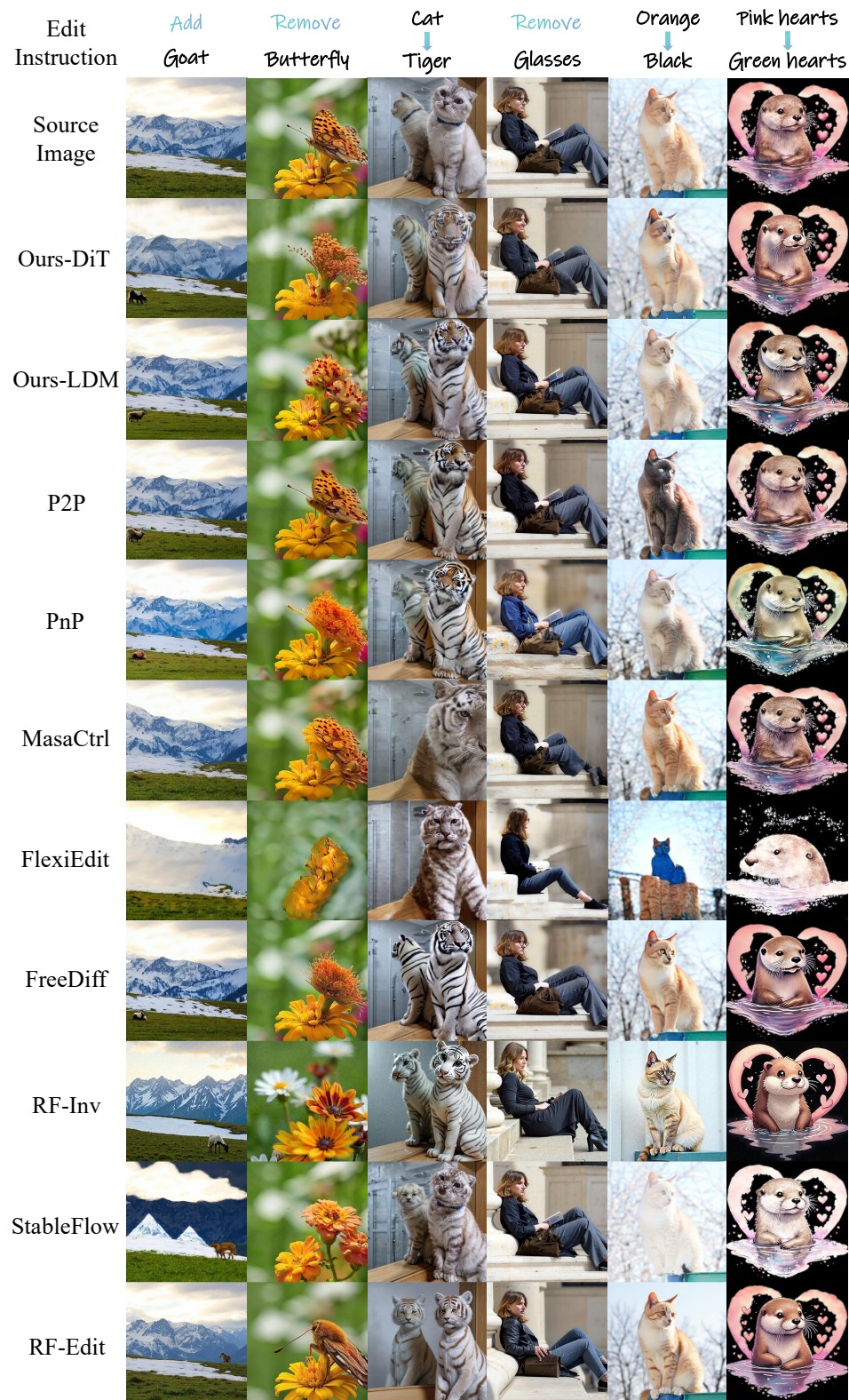

Figure 12: Additional qualitative editing results on PIE-Bench Part II. The rightmost three columns show failure cases from our main paper. Competing methods often fail to preserve fine details or to perform the desired semantic edits (*e.g.*, removing woman's glasses, changing the cat's color).

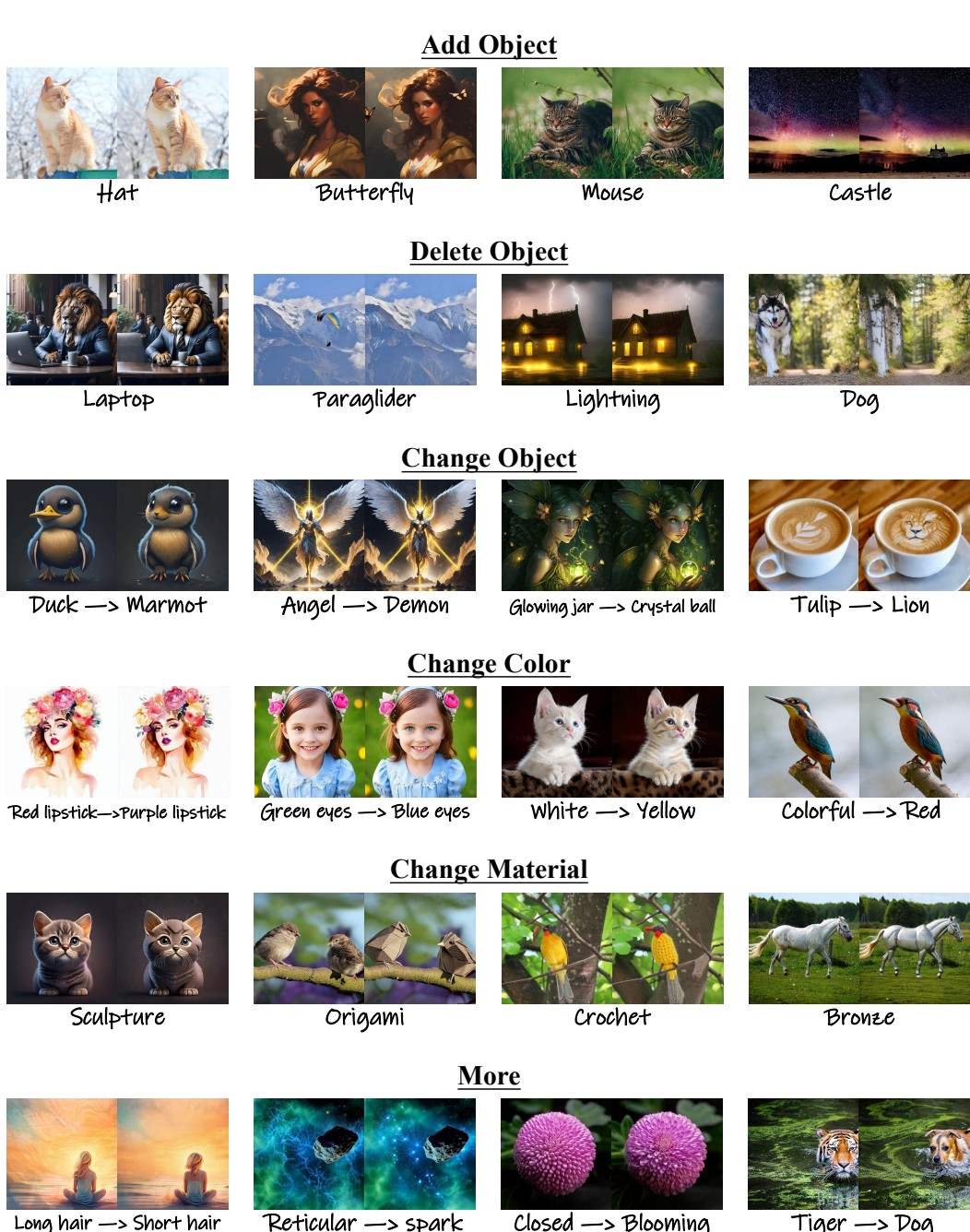

Figure 13: Additional qualitative results on diverse editing types from PIE-Bench Part I.

Figure 14: Additional qualitative results on diverse editing types from PIE-Bench Part II.

