# OpenReview forum: "FSI-Edit: Frequency and Stochasticity Injection for Flexible Diffusion-Based Image Editing"
_NeurIPS.cc/2025/Conference — NeurIPS 2025 poster_

### Official Review · Reviewer_MHta · 2025-06-14

**Clarity:** 3
**Significance:** 3
**Originality:** 3
**Rating:** 4
**Confidence:** 2

**Summary:**

This paper aims to address a critical issue in text-to-image (T2I) editing based on Latent Diffusion Models (LDMs): how to enable flexible image editing without destroying the original image content. The study seeks to overcome the limitations of existing LDM-based T2I editing methods through frequency and randomness injection.

**Questions:**

See Weaknesses

**Ethical Concerns:**

["NO or VERY MINOR ethics concerns only"]

**Final Justification:**

I'm very sorry, but I really can't understand this paper. Please refer to the comments of other reviewers.

**Limitations:**

yes

**Quality:**

3

**Strengths And Weaknesses:**

Advantages:
1. The paper is fluently written with clear expressions, and the figures are highly appealing.
2. The demonstrated effects outperform comparative methods both qualitatively and quantitatively.

Disadvantages:
1. As I have limited understanding of this field and am an outsider, I still find it difficult to comprehend the paper thoroughly even after careful reading, making it challenging to provide effective suggestions. It is recommended to refer more to the opinions of other professional reviewers. Apologies for the inconvenience.

---

> ### Author Rebuttal · Authors · 2025-07-31
>
> We sincerely thank the reviewer for their time and effort in reading our paper and providing feedback, especially from the perspective of a broader audience. We appreciate your thoughtful comments, and we will take this as a valuable reminder to further improve the clarity and accessibility of our presentation for readers from diverse backgrounds.

---

### Official Review · Reviewer_7PE5 · 2025-06-18

**Clarity:** 2
**Significance:** 3
**Originality:** 3
**Rating:** 5
**Confidence:** 3

**Summary:**

The authors propose a method that allows both rigid and non-rigid image editing using diffusion models. The method is based on introducing high frequency part of the reconstruction features into the generation features along with some noise.

**Questions:**

- It is not clear what is the motivation behind Inversion Trajectory Navigation (ITF). Please add a few sentences to discuss the motivation behind including this component in your proposed framework.
- Please provide the visual ablation of FRF in figure 6.

**Ethical Concerns:**

["NO or VERY MINOR ethics concerns only"]

**Final Justification:**

After reading author's rebuttal I have increased my score from weak accept (4) to accept (5)

**Limitations:**

- According to equation (1), and (2), you multiply the low frequency component of $v$ by $alpha$ and then you add $v$ again after the inverse FFT. This basically boosts the low frequency part of $v$ by a factor of $\alpha$. Please explain the reasoning behind this design. Also adding back $v$ in equation (2) basically keeps all the high frequency components of $v$. So if there is a clash between the high frequency parts of the edited image and the source your method will fail. This failure is also evident in the teaser figure (rigid editing - painting -> photo - bunny and cat image) where the edited image still looks like a painting and not a photograph.

**Quality:**

3

**Strengths And Weaknesses:**

Strengths:
- The proposed method enables faithful editing for both rigid and non-rigid cases in a single unified framework.
- Extensive quantitative evaluation with many different image-editing methods.
- The qualitative results demonstrate that the proposed method outperforms the existing baselines.
- The quantitative results are mixed but they show clear improvement for non-rigid editing and for the transformer architecture.


Weaknesses:
- The proposed method is complicated and involves many components with many hyperparameters to tune.
- ITN only slightly improves the quantitative results. according to Table 3 (main). The authors also don't provide any motivations for including it in their proposed model.
- The quantitative results are mixed. The results on PIE-Bench, Table 1, show marginal or no improvement compared to the existing method. A user study is probably needed to obtain a more decisive quantitative evaluation.

---

> ### Author Rebuttal · Authors · 2025-07-31
>
> We sincerely thank the reviewer for the detailed and thoughtful feedback. We greatly appreciate the recognition of our unified framework's effectiveness across both rigid and non-rigid editing tasks, as well as the positive assessment of our extensive experiments and superior qualitative results.
> We also acknowledge the reviewer’s concerns regarding the model’s complexity, the marginal gains of ITN in quantitative metrics, and the design rationale behind frequency manipulation. Below, we provide clarifications on these aspects and address all specific questions and limitations point by point.
>
> **[W1: Complicated methods, and many hyperparameters]**
> We understand the concern about complexity. Our method is indeed structurally enriched to handle challenging non-rigid editing, but it remains practical: it runs with moderate memory ($16.08 GB$) and reasonable inference time ($21.38 s$).
> Importantly, our ablation studies in Appendix D show that performance is robust to a wide range of hyperparameter settings, meaning users can directly adopt our recommended defaults without tuning. Thus, while our method involves several components, it remains efficient, robust, and easy to use.
>
> **[W2&Q1: The motivation of ITN]**
> The motivation for Inversion Trajectory Navigation (ITN) is to improve DDIM inversion fidelity by preventing information loss across timesteps. Inspired by our successful FRF module, ITN applies frequency fusion to the temporal domain. It enriches the current latent by incorporating stable, low-frequency structural cues from a previous, cleaner timestep, while retaining current high-frequency details. This creates a more robust trajectory for more faithful reconstructions, as validated by our ablation studies (Main Paper Table 3, Figure 6a).
>
> **[W3: Mixed quantitative results and User study]**
> Thank you for your supportive review and for your insightful feedback. We particularly appreciate your comment on the quantitative results and your excellent suggestion that a user study would provide a more decisive evaluation.
>
> 1.**User Study Confirms Superior Performance.**
> We completely agree with your assessment and are pleased to report that we have followed your advice and conducted a formal user study. For this study, we used approximately 20 cases from PIE-Bench. Participants were asked to evaluate results from our method and key baselines on a scale of 0-100 across two critical axes: (1) **semantic alignment** with the text prompt and (2) **background consistency**. To prevent bias, the order in which the results were presented was randomized. From this process, we collected 11 valid responses, with an average completion time of 20 minutes. The detailed results are summarized below:
>
> Table 1: User Study
> |Method|Editing (Semantic)|Background|
> |-|-|-|
> |PnP|34.58|73.35
> |MasaCtrl|35.68|72.05|
> |FlexiEdit|39.55|64.60|
> |FreeDiff|36.69|73.04|
> |RF-Inv|55.86|63.09|
> |StableFlow|44.78|71.80|
> |RF-Edit|60.75|67.88|
> |Ours-DiT|**78.55**|**80.66**|
>
> The results confirmed that our method was consistently rated as the best overall, achieving a superior, perceptually preferred balance between semantic accuracy and background preservation. We are committed to including an expanded version of this study, with more participants and diverse editing tasks, in the final paper.
>
> 2.**A Holistic View on Quantitative Metrics.**
> This user study helps to contextualize why the quantitative metrics in Main Paper Table 1 might appear mixed at first glance. Image editing evaluation involves an inherent trade-off between semantic fidelity and background preservation. Often, methods that excel on a single preservation metric do so by under-editing, making only minimal changes and thus failing to meet the user's intent. For example, methods like P2P or StableFlow score very high on background preservation but low on semantic similarity.
>
> Our goal was never to maximize a single metric at the expense of the edit's actual quality, but to achieve the most effective and balanced result. To provide a clearer quantitative picture that reflects this balance, we calculated the average rank of each method across all metrics. This holistic analysis smooths out the effects of the trade-off and reveals the most robust method overall.
>
> Table 2: Average Ranking for General Editing (Main Paper Table 1)
> |Method|Avg Rank|Final Rank|
> |-|-|-|
> |P2P|3.71|2|
> |PnP|7.43|8|
> |MasaCtrl|8.71|10|
> |FlexiEdit|4.29|4|
> |FreeDiff|6.29|6|
> |Ours-LDM|5.00|5|
> |RF-Inv|9.43|11|
> |StableFlow|6.86|7|
> |RF-Edit|7.57|9|
> |DCEdit|4.14|3|
> |**Ours-DiT**|**2.43**|**1**|
>
> Table 3: Average Ranking for Non-rigid Editing (Main Paper Table 2)
> |Method|Avg. Rank|Final Rank|
> |-|-|-|
> |PnP|5.43|6|
> |MasaCtrl|5.71|7|
> |FlexiEdit|8.57|9|
> |FreeDiff|3.14|2|
> |Ours-LDM|3.71|3|
> |RF-Inv|6.57|8|
> |StableFlow|4.57|4|
> |RF-Edit|4.71|5|
> |Ours-DiT|**2.57**|**1**|
>
> As shown, when performance is viewed holistically, our DiT-based method consistently achieves the 1st overall rank, demonstrating its superior, well-rounded performance. We hope that the positive results of the new user study, combined with this more holistic perspective on our quantitative results, have addressed your concerns. Thank you again for your valuable and constructive feedback which has helped us strengthen the evaluation of our work.
>
> **[Q2: Visual ablation of FRF]**
> Thank you for this request. While rebuttal stage limitations prevent us from including new figures, we will certainly add detailed visual ablations to the final paper.
>
> To describe the key visual difference, the ‘bird-to-X’ non-rigid transformation is the most striking example. Without FRF, the model exhibits a clear failure of structural adaptation. The bottom-right quadrant of the object retains the shape of the original bird's body, failing to form the lower-right ‘leg’ of the target ‘X’. This results in a malformed, hybrid object. This specific failure demonstrates that FRF is crucial for mitigating this source structure inertia and enabling the complete, coherent transformations that are central to our method's success.
>
> **[L1.1: Reason of FRF]**
> Thank you for this excellent question regarding the design of our Frequency-aware Residual Fusion (FRF). The formulation is deliberately designed to achieve a principled fusion of features from the source ($u$) and target ($v$), which we will break down below.
>
> Our core objective is to enrich the target feature $v$ with high-fidelity details from $u$, without corrupting the essential structure of $v$ that guides the generation.
>
> **Equation (1): Fusing Structure and Detail in the Frequency Domain.**
> This step combines the best of both features in the spectral domain.
> $\mathcal{L}$ is used to isolate the **low-frequency** components of the target $v$. These represent the global structure, layout, and overall shape—the essential "scaffolding" of the target.
> $\mathcal{H}$ is used to extract the **high-frequency** components of the source $u$. These correspond to fine-grained textural details, edges, and patterns. By adding these, $\mathcal{F}_{fuse}$ becomes a hybrid feature that contains the structural backbone of the target $v$ and the textural richness of the source $u$.
>
> **Equation (2): Reinforcement and Stabilization via Residual Connection.**
> The second step, adding back the original $v$, is equally critical. This residual connection serves two vital purposes:
> **Reinforcing Semantic Integrity**: It ensures the final output remains strongly anchored to the target $v$'s complete structure, preventing the high-frequency details from $u$ from overpowering it.
> **Mitigating Artifacts**: It helps correct for potential phase misalignments or other artifacts that can arise from manipulating signals purely in the frequency domain, resulting in a cleaner and more stable feature map.
> In essence, this two-step process allows our model to perform a sophisticated transplant of detail while rigorously preserving the target's core shape and identity.
>
> **[L1.2: Limitation]**
>     Thank you for this insightful observation. The reviewer is entirely correct, and the failure case in our teaser figure (the painting-to-photo edit) is a perfect illustration of this inherent challenge. When the high-frequency texture of the source (e.g., paint strokes) clashes with the semantics of the target prompt (e.g., a photograph), a conflict arises.
>
> This highlights the deliberate synergy between our FRF and SNI modules. While FRF is designed to transfer rich, high-frequency details, this carries an inherent risk of introducing an overly strong texture bias. The another role of the SNI module is to counteract this effect. By injecting stochasticity, it provides the model with the flexibility to deviate from the source texture and better align with the target semantics. However, it's not always perfectly resolved.
>
> This feature clash is a fundamental challenge for most tuning-free editing methods that rely on direct feature fusion. Preserving source identity via its features will always carry a risk of preserving unwanted source attributes like style or texture.
> While SNI serves as an effective mitigation, we agree that a more principled solution is a valuable direction. For future work, we plan to explore a semantically-guided fusion mechanism. For example, one could use attention maps to dynamically weigh the high-frequency fusion, reducing its influence in regions where a complete style change is required.

---

> ### Comment · Reviewer_7PE5 · 2025-08-03
>
> Thanks for doing a user study during the short rebuttal period. My concerns are mostly resolved.
>
> If accepted, for the final version of the paper please repeat your user study with more users and include confidence intervals when reporting your user study results.
>
> I have increased my score from 4 to 5.

---

> > ### Author Response · Authors · 2025-08-03
> >
> > Thank you very much for your encouraging feedback. We truly appreciate your suggestion. We will repeat the user study with more participants and include confidence intervals in the final version of the paper.

---

### Official Review · Reviewer_Xe12 · 2025-06-29

**Clarity:** 4
**Significance:** 2
**Originality:** 2
**Rating:** 4
**Confidence:** 3

**Summary:**

This paper proposes an image editing framework, i.e., FSI-Edit, designed for both rigid and non-rigid editing tasks in diffusion-based image generation. The proposed method introduces two main mechanisms: Frequency Residual Fusion (FRF) for high-frequency detail injection and Stochastic Noise Injection (SNI) to enhance generative flexibility. The expensive experiences are conducted on the PIE-Bench dataset.

**Questions:**

See the strengths and weaknesses

**Ethical Concerns:**

["NO or VERY MINOR ethics concerns only"]

**Final Justification:**

Thanks for the rebuttal. Most of my concerns have been addressed.

**Limitations:**

See the strengths and weaknesses

**Quality:**

3

**Strengths And Weaknesses:**

Strengths:

1.	The proposed method is conceptually well-motivated and integrates frequency-domain information and stochasticity.

2.	The paper is generally well-organized and clearly written. Figures help to convey the methodology effectively.

Weaknesses:

1.	While the method incorporates some novel ideas, such as frequency-based feature fusion in attention layers, however, similar principles have been explored in recent works like FlexiEdit and FreeDiff.

2.	The overall comparisons, especially among UNet-based methods, most of evaluation metrics show a noticeable gap between the proposed method and state-of-the-art (SOTA) baselines. This weakens the claim of superiority of the proposed approach.

3.	The ablative results (Table 3) indicate that both "w/o FRF" and "w/o SNI" variants outperform the full model in most of metrics, which fails to clearly demonstrate the contribution of each component.

4.	It is not clear whether the limitations mentioned also occur in other methods. A more comprehensive and critical analysis is recommended.

---

> ### Author Rebuttal · Authors · 2025-07-31
>
> We sincerely thank you for the constructive feedback and for recognizing the conceptual novelty and clarity of our proposed method. While we appreciate the positive remarks regarding our framework and its presentation, we would like to clarify several misunderstandings in the evaluation and address the concerns regarding the experimental comparisons, component effectiveness, and related work overlap. Below, we provide point-by-point responses to each comment.
>
> **[W1: Difference between FlexiEdit and FreeDiff]**
> Thank you for acknowledging the novelty of our frequency-based feature fusion. We appreciate the opportunity to clarify the key distinctions between our method and prior works like FlexiEdit and FreeDiff. While the concept of frequency-domain manipulation has appeared in prior works, the way we leverage it is fundamentally different in both purpose and implementation. **FlexiEdit** applies frequency filtering at the latent noise level, which is early in the generation process and lacks semantic alignment with intermediate representations. **FreeDiff** operates on classifier-free guidance (CFG) signals using frequency-based constraints, without direct interaction with model features.
>
> In contrast, our Frequency Residual Fusion (FRF) is, to our knowledge, the first to systematically inject frequency-domain manipulation directly into **deep feature maps** during the generation process. This design specifically targets the challenge of non-rigid semantic editing, which requires complex, large-scale structural changes (e.g., turning a bird into X shape). Our approach facilitates more controllable and precise edits by integrating high-frequency details from the source while preserving low-frequency semantics from the target. Visualizations and experiment results in the paper and supplementary materials demonstrate the superior effect of this design in such challenging tasks.
>
> Furthermore, our other components, Stochastic Noise Injection (SNI) and Inversion Trajectory Navigation (ITN), address distinct aspects of controllability and reconstruction quality. These ideas are not explored in the aforementioned works, further differentiating our method.
>
> Thus, while frequency-domain ideas are not entirely new, our framework offers a novel, integrated, and effective way to apply them, particularly for flexible and large-scale image manipulation.
>
> **[W2: Gap in comparison with SOTA]**
> We respectfully disagree with the characterization that a gap on specific metrics weakens our claim of superiority. In fact, this phenomenon highlights a critical misunderstanding of the evaluation of image editing. The task is not about maximizing any single metric, but about achieving an optimal **trade-off** between **semantic alignment** and **background preservation**.
>
> 1.**The Inherent Trade-Off of Image Editing.**
> Excelling at one type of metric often signifies failure in the other, leading to two undesirable extremes. **Under-editing**: Methods that rank SOTA on preservation metrics (e.g., LPIPS) often do so by failing to meaningfully alter the image according to the prompt. This results in `high background consistency` but `poor semantic alignment`. In our paper, baselines like P2P, FreeEdit, and StableFlow exemplify this, achieving top preservation scores at the cost of insufficient editing.
> **Over-editing**: Conversely, a method that perfectly matches the text prompt but destroys the original background and structure is no longer performing an edit, but rather an unconstrained text-to-image generation. Hence, `high semantic alignment` but `poor background consistency`.
>
> A truly superior editing method must navigate this trade-off effectively. Our approach is designed to find this balance, delivering strong semantic changes while respecting the original image's integrity, as our qualitative results demonstrate.
>
> 2.**Holistic Quantitative Analysis.**
> To provide a fair, holistic comparison that honors this trade-off, we analyzed the average rank of each method across all $7$ evaluation metrics from our main paper. This approach normalizes the performance across competing objectives and reveals the most well-balanced method overall. The results below unequivocally demonstrate that our method is SOTA.
>
> Table 1: Average Ranking for General Editing (Main Paper Table 1)
> |Method|Avg Rank|Final Rank|
> |-|-|-|
> |P2P|3.71|2|
> |PnP|7.43|8|
> |MasaCtrl|8.71|10|
> |FlexiEdit|4.29|4|
> |FreeDiff|6.29|6|
> |Ours-LDM|5.00|5|
> |RF-Inv|9.43|11|
> |StableFlow|6.86|7|
> |RF-Edit|7.57|9|
> |DCEdit|4.14|3|
> |Ours-DiT|**2.43**|**1**|
>
> Table 2: Average Ranking for Non-rigid Editing (Main Paper Table 2)
> |Method|Avg. Rank|Final Rank|
> |-|-|-|
> |PnP|5.43|6|
> |MasaCtrl|5.71|7|
> |FlexiEdit|8.57|9|
> |FreeDiff|3.14|2|
> |Ours-LDM|3.71|3|
> |RF-Inv|6.57|8|
> |StableFlow|4.57|4|
> |RF-Edit|4.71|5|
> |Ours-DiT|**2.57**|**1**|
>
> A crucial point highlighted by these tables is the consistent effectiveness of our method across different backbones. While `Ours-DiT` establishes the new state-of-the-art, our UNet variant also achieves a top-tier rank (#3 in non-rigid editing). This proves that our method significantly enhances the standard UNet architecture, while also unlocking the full potential of the more powerful DiT backbone. This versatility further strengthens our claims.
>
> 3.**Qualitative and User Study Validation.**
> This quantitative superiority is further supported by our qualitative results. To formalize this, we conducted a user study to assess human perceptual preferences. We presented 11 participants with about 20 randomized image sets from PIE-Bench and asked them to score the results based on both semantic alignment with the prompt and background consistency. The results confirmed that our method was judged to be the best overall, achieving the most favorable balance of successfully editing the subject while preserving the background.
>
> Table 3: User Study
> |Method|Editing (Semantic)|Background|
> |-|-|-|
> |PnP|34.58|73.35
> |MasaCtrl|35.68|72.05|
> |FlexiEdit|39.55|64.60|
> |FreeDiff|36.69|73.04|
> |RF-Inv|55.86|63.09|
> |StableFlow|44.78|71.80|
> |RF-Edit|60.75|67.88|
> |**Ours-DiT**|**78.55**|**80.66**|
>
> Therefore, our quantitative, qualitative, and human-centric results robustly support our claim of superiority.
>
> **[W3: Ablation variants outperform full model]**
> Thank you for the comment. The ablation results in Main Paper Table 3 can validate each component when interpreted correctly for the **non-rigid editing task**. For this task, an excessively high preservation score (e.g., LPIPS) signifies editing failure (under-editing), not superior performance, as it indicates the model failed to make the required structural changes.
> Our ablation results clearly demonstrate this trade-off:
> - **w/o FRF**: Its top preservation score and lowest semantic alignment prove that FRF is essential to enable structural deformation. Without it, the model simply resists the edit.
> - **w/o SNI**: Its high preservation and weak alignment show that SNI is crucial for providing the flexibility needed to find a successful edit path.
> - **Full Model**: It achieves the best semantic alignment with a necessary and expected drop in preservation—the signature of a successful and significant non-rigid edit.
>
> In summary, the study confirms via proof by contradiction that both FRF and SNI are indispensable. Their synergy is essential for performing the challenging, flexible edits that are the core goal of our work.
>
> **[W4: Failure cases analysis]**
> Thank you for the comment. We should point out that the comprehensive analysis is already provided in our paper. The **last three columns of Figure 11 in the Appendix** are dedicated to this exact head-to-head comparison. These results make it clear that the identified limitations are not weaknesses of our work, but are systemic challenges faced by all current state-of-the-art methods.
>
> To summarize the findings from that comparison:
> - **Removing glasses**: Most methods either fail entirely or introduce severe artifacts. In contrast, our approach provides a more visually coherent result.
> - **Color-specific edits**: Most methods struggle with fine-grained color manipulation, often failing to correctly identify the target object before applying the specified color.
>
> Therefore, this direct comparison demonstrates that while most of all methods fail, our approach exhibits stronger and more consistent performance than others on these editing tasks.

---

> > ### Comment · Reviewer_Xe12 · 2025-08-03
> >
> > Thanks for the rebuttal. However, I am still have following concerns with regard to my previous comments.
> >
> > 1. Regarding W2, I am curious about the average ranking for the UNet-based and Transformer-based methods.
> >
> > 2. For W3, the statement "FRF is essential for enabling structural deformation", which should be supported by the "w FRF" results, rather than "w/o FRF." The same applies to SNI.
> >
> > 3. For W4, where is Figure 11? I cannot find it in the manuscript or Appendix.

---

> > > ### Author Response · Authors · 2025-08-03
> > >
> > > Thank you very much for your follow-up questions and for carefully reviewing our rebuttal. We sincerely appreciate the opportunity to clarify these remaining points in detail. Below we address your concerns regarding W2, W3, and W4 point by point.
> > >
> > > **W2: Average ranking for the UNet-based and Transformer-based methods.**
> > >
> > > We have computed the average rankings separately for UNet-based and Transformer-based methods from the data in Table 1 of the main paper. The results are summarized below:
> > >
> > > Table 1: Main Paper Table 1 for UNet
> > > |Method|Avg Rank|Final Rank|
> > > |-|-|-|
> > > |P2P|2.14|1|
> > > |PnP|4.57|5|
> > > |MasaCtrl|5.29|6|
> > > |FlexiEdit|2.57|2|
> > > |FreeDiff|3.71|4|
> > > |Ours-LDM|2.71|3|
> > >
> > > Table 2: Main Paper Table 1 for Transformer
> > > |Method|Avg Rank|Final Rank|
> > > |-|-|-|
> > > |RF-Inv|4.43|5|
> > > |StableFlow|3.14|3|
> > > |RF-Edit|3.57|4|
> > > |DCEdit|2.29|2|
> > > |Ours-DiT|1.43|1|
> > >
> > > Similarly, for the evaluation in Table 2 of the main paper, we summarize the average ranks in:
> > >
> > > Table 3: Main Paper Table 2 for UNet
> > > |Method|Avg Rank|Final Rank|
> > > |-|-|-|
> > > |PnP|3.43|4|
> > > |MasaCtrl|3.00|3|
> > > |FlexiEdit|4.86|5|
> > > |FreeDiff|1.71|1|
> > > |Ours-LDM|2.00|2|
> > >
> > > Table 4: Main Paper Table 2 for Transformer
> > > |Method|Avg Rank|Final Rank|
> > > |-|-|-|
> > > |RF-Inv|3.43|4|
> > > |StableFlow|2.43|2|
> > > |RF-Edit|2.71|3|
> > > |Ours-DiT|1.43|1|
> > >
> > > **W3: w/o FRF and w/o SNI.**
> > >
> > > Thank you for the thoughtful comment. In our ablation setting, we start from the Full model, which already demonstrates strong non-rigid editing capability. The effectiveness of each component is thus reflected in the performance degradation when that component is removed.
> > >
> > > Specifically, the w/o FRF result shows that removing FRF leads to a failure in structural deformation, indicating that FRF plays a crucial role in enabling it. Similarly, the drop in semantic alignment when SNI is removed validates its role in guiding the editing with stochasticity. Therefore, the negative impact of removal in a full-model context provides empirical support for the necessity of each module. We hope this clarifies our reasoning.
> > >
> > > **W4: Clarification on Figure 11.**
> > >
> > > We sincerely apologize for the confusion caused by the incorrect figure reference in our previous response. The intended figure is **Figure 4 in the Appendix**, titled `Additional qualitative editing results on PIE-Bench Part II.` The rightmost three columns of Figure 4 illustrate failure cases across different methods.

---

> > > > ### Comment · Reviewer_Xe12 · 2025-08-05
> > > >
> > > > Thanks for the persuasive rebuttals. Most of my concerns have been adequately addressed. I would like to increase my score.

---

> > > > > ### Author Response · Authors · 2025-08-05
> > > > >
> > > > > We sincerely thank you for the constructive and thoughtful feedback, as well as for the time and effort devoted to evaluating our work and helping us improve the clarity and quality of the manuscript.

---

### Official Review · Reviewer_9JHL · 2025-07-03

**Clarity:** 2
**Significance:** 2
**Originality:** 3
**Rating:** 4
**Confidence:** 4

**Summary:**

FSI-Edit introduces a novel diffusion-based image editing framework that enhances flexibility in both rigid and non-rigid edits. While traditional methods often struggle with semantic mismatches and limited generative diversity due to direct feature replacement, FSI-Edit effectively alleviates this drawbacks through two key mechanisms: Frequency Residual Fusion (FRF), which injects only high-frequency details from the source to preserve texture while avoiding structural conflicts, and Stochastic Noise Injection (SNI), which introduces controlled randomness to unlock diverse transformations. Extensive experiments show that FSI-Edit outperforms existing methods in terms of both semantic alignment and content preservation.

**Questions:**

Q1. For further questions and concerns, please refer to the Major and Minor Weaknesses sections above.

**Ethical Concerns:**

["NO or VERY MINOR ethics concerns only"]

**Final Justification:**

During the rebuttal, my major concerns regarding (1) the effectiveness and necessity of each proposed component, (2) computational cost, and (3) comparison with additional baselines were resolved. In addition, the authors provided a detailed motivation and formulation for stochastic noise injection (SNI), and discussed improvements in paper writing and presentation, which further alleviated my concerns. Therefore, I have increased my score to 4.

**Limitations:**

Yes

**Quality:**

3

**Strengths And Weaknesses:**

**Strengths**

S1. This paper analyzes the drawbacks of the prior image editing method, especially for non-rigid edits, and effectively alleviates the problem by proposing the FSI-Edit framework. Furthermore, using frequency-domain features for image editing is both novel and impressive.

S2. This paper proposes three novel components; 1) Frequency Residual Fusion, 2) Stochastic Noise Injection, and 3) Inversion Trajectory Navigation to enable flexible and complex translations for the image.

S3. The authors provide various qualitative results in the main and the supplementary material, including non-rigid editing tasks. Moreover, by evaluating the framework using two distinct generative backbones (LDM [1] and DiT [2]), the authors demonstrate the robustness and generalizability of the proposed approach across different backbone architecture.

S4. The paper includes comprehensive quantitative comparison against a diverse set of baseline methods. FST-Edit consistently shows superior performance across prior works, which strongly supports the effectiveness of the proposed work.

**Major Weakness**

W1. The ablation study results (including those in Appendix Sec. D) are not compelling. It raises the question of whether every component is necessary. While the differences in CLIP similarity are minor, performance in background & structure preservation varies significantly (especially w/o FRF and SNI). A more detailed investigation into the effect of each component would strengthen the foundation of the paper. Furthermore, it would be highly beneficial if the authors could provide a computational overhead (including GPU memory consumption and runtime) introduced by each component, to better quantify the trade-off in efficiency. This is my major concern with this paper.

W2. For ITN, it lacks comparison with reconstruction-refining methods, such as Null-text inversion or edit friendly inversion.
Effectiveness of Inversion Trajectory Navigation (ITN) as a latent-space refinement strategy is not compared against other established reconstruction-refinement techniques, such as Null-text inversion [3] or Edit-Friendly inversion [4]. A direct comparison with these methods would provide more clarity on the advantage of ITN for source image reconstruction.


**Minor Weakness**

W3. Is there any basis that stochastic noise injection on query and key in self-attention map makes sense and derives a good result? While the authors reference stochastic DDIM [5, 6] for the motivation, it’s weak since in stochastic DDIM, noise injection into the latent variable is mathematically grounded via SDE [7] (and Langevin MCMC). In contrast, injecting noise into attention maps lacks such theoretical justifications and it seems quite ambiguous.

W4. The authors wrote preliminaries and details in the supplementary material. I appreciate the detailed discussion of those. In the final version, moving some of these sections (e.g. preliminary, ablation study) into the main paper could improve overall clarity and reader comprehension.

References

[1] Rombach, Robin, et al. "High-resolution image synthesis with latent diffusion models." in CVPR (2022).

[2] Peebles, William, and Saining Xie. "Scalable diffusion models with transformers." in ICCV (2023).

[3] Mokady, Ron, et al. "Null-text inversion for editing real images using guided diffusion models." in CVPR (2023).

[4] Huberman-Spiegelglas, Inbar, Vladimir Kulikov, and Tomer Michaeli. "An edit friendly ddpm noise space: Inversion and manipulations." in CVPR (2024).

[5] Ho, Jonathan, Ajay Jain, and Pieter Abbeel. "Denoising diffusion probabilistic models." in NeurIPS (2020).

[6] Song, Jiaming, Chenlin Meng, and Stefano Ermon. "Denoising diffusion implicit models." in ICLR (2021).

[7] Song, Yang, et al. "Score-based generative modeling through stochastic differential equations." in ICLR (2021).

---

> ### Author Rebuttal · Authors · 2025-07-31
>
> We sincerely thank the reviewer for the thorough and constructive feedback. We appreciate the recognition of our contributions, including the introduction of a novel frequency-domain editing framework, and the strong empirical performance across diverse settings. Your positive assessment of our method's design, robustness, and evaluation is highly encouraging. Below, we provide point-by-point responses to address the concerns and clarify potential misunderstandings.
>
> **[W1.1 The necessity of each component]**
> We wish to highlight that evaluating ablation studies for **non-rigid editing** using preservation metrics can be inherently challenging and potentially misleading. This is a recognized difficulty in the field. For instance, non-rigid editing methods, MasaCtrl (ICCV23) and FlexiEdit (ECCV24), primarily rely on qualitative visual comparisons for their ablation studies rather than quantitative metrics. Following this established practice, we will include detailed visual ablations in the final version to more effectively demonstrate the contribution of each component.
>
> Nevertheless, we respectfully argue that our current quantitative results are also compelling once they are interpreted correctly for this challenging task. The core of the issue lies in how to define success when significant structural changes are required.
>
> First, we establish the correct framework for interpretation. **Under-Editing Trap.** It is often easy to achieve a deceptively high CLIP score by under-editing (Our Baseline, Table1 below). This is because the target prompt frequently contains descriptions of the background or other non-edited elements (e.g., a blue cat sitting-> a blue cat standing). A model that fails to edit the subject will still get credit for the correctly rendered un-edited regions. **Metric Scaling Trap.** Our preservation metrics are scaled by large factors (e.g., LPIPS×10³). This makes numerical differences appear much larger than their actual visual impact.
>
> Table 1: More Ablation Study
> |Method|Distance×10³↓|PSNR↑|LPIPS×10³↓|MSE×10⁴↓|SSIM×10²↑|CLIP Whole↑|CLIP Edited↑|
> |-|-|-|-|-|-|-|-|
> |Baseline|8.45|27.97|68.94|24.93|87.80|24.71|20.91|
> |Base-X|96.66|56.54|8.08|0.02|99.14|19.74|19.80|
> |Ours-X|238.81|36.50|61.66|2.24|94.80|29.60|30.62|
> |Base-add|1.81|27.38|50.32|18.26|78.96|22.13|14.93|
> |Ours-add|17.29|22.69|139.98|53.83|68.96|25.82|25.83|
>
> To illustrate this principle, we compare against the baseline on two representative non-rigid edits from our paper.
> - **Bird-to-X (Fig5 Row3).** The base result is still a blue bird. It achieves near-perfect preservation scores precisely because it completely fails to perform the edit. In contrast, ours successfully executes the edit. The associated "worsening" of preservation scores is direct proof that the required drastic structural change has actually occurred.
> - **Adding a Person to an Elephant (Fig5 Row6).** The base barely changes the image and consequently fails the semantic task. Ours succeeds. This semantic improvement is correctly accompanied by a significant change in the image structure, which the task demanded.
>
> This not only demonstrates the effectiveness of our overall framework but also proves that its components are necessary to solve non-rigid editing.
>
> **[W1.2: Computational overhead of each module]**
> We first report the resource consumption and runtime of the baseline methods. GPU memory usage and runtime are measured for processing a single image using our method (FSI-Edit-DiT) and several open-source baselines. All methods are run on a single NVIDIA RTX 4090 GPU, except for RF-Inv, StableFlow, and RF-Edit, which are executed on an A100 GPU due to their higher memory demands. Runtime is averaged over 10 images to ensure stable measurement.
>
> Table 2: GPU Memory Usage and Inference Time (Per Image)
> |Method|P2P|PnP|MasaCtrl|FlexiEdit|FreeDiff|RF-Inv|StableFlow|RF-Edit|Ours (DiT)|
> |-|-|-|-|-|-|-|-|-|-|
> | **GPU (GB)** |10.95|8.99|11.42|18.73|6.08|69.22|35.39|32.91|16.08|
> |**Time (s)**|34.84|18.09|21.71|38.97|17.41|76.74|26.07|34.51|21.38|
>
> The frequency-domain operations do not introduce significant additional GPU overhead, so the overall GPU memory usage remains around $16.08 GB$. In terms of runtime, the baseline requires $17.19s$. The SNI module introduces a negligible overhead of $0.73s$, as it only injects noise into the intermediate features. The ITN module, applied during the inversion stage over 50 steps, adds $0.31s$. The FRF module is applied to the Q and K features across 13 self-attention layers and runs for 25 denoising steps, contributing $3.15s$ to the total runtime. It shows that our method achieves a well-balanced trade-off between computational efficiency and editing effectiveness.
>
> **[W2: Direct Comparison for ITN]**
> We conducted a dedicated experiment based on the P2P backbone, comparing three representative inversion approaches: Direct Inversion (Direct) [1], Null-text Inversion (NT) [2], and Edit-Friendly Inversion (Friend) [3]. We evaluate these methods from two perspectives: **editing image quality**, and **source image reconstruction quality**. The corresponding results are reported in the following tables:
>
> Table 3: Editing Performance Comparison
> |Method|Distance×10³↓|PSNR↑|LPIPS×10³↓|MSE×10⁴↓|SSIM×10²↑|CLIP Whole↑|CLIP Edited↑|
> |-|-|-|-|-|-|-|-|
> |DDIM|69.43|17.87|208.80|219.88|71.56|25.01|**22.44**|
> |Direct|11.65|**27.22**|**54.55**|**32.86**|**85.10**|25.02|22.10|
> |NT|13.44|27.07|59.88|35.47|84.55|24.75|21.87|
> |Friend|11.07|26.17|58.73|38.21|84.26|25.22|22.13|
> |**ITN(Ours)**|**10.31**|26.34|57.57|37.20|84.60|**25.35**|22.13|
>
> Table 4: Source Image Reconstruction Quality
> |Method|Distance×10³↓|PSNR↑|LPIPS×10³↓|MSE×10⁴↓|SSIM×10²↑|CLIP Whole↑|
> |-|-|-|-|-|-|-|
> |DDIM|70.23|17.76|210.84|224.43|71.39|27.07|
> |Direct|2.95|**30.57**|**31.41**|**17.60**|**87.50**|25.45|
> |NT|3.30|30.17|33.50|18.94|87.13|25.50|
> |Friend|3.12|30.37|32.66|18.21|87.19|25.55|
> |**ITN(Ours)**|**2.33**|30.49|31.88|17.85|87.47|**25.64**|
>
> As shown, our ITN achieves the lowest distance in both editing and reconstruction settings, and performs best in CLIP similarity. While Direct Inversion slightly outperforms in other background metrics. These results confirm the benefit of our ITN design, which leverages trajectory-aware frequency fusion between latent states to improve semantic coherence and source fidelity. ITN not only ensures **high-quality source reconstruction** but also facilitates **flexible and meaningful image edits**.
>
> **[W3: Stochastic Noise Injection into Query/Key]**
> Thank you for this insightful question. Our stochastic noise injection (SNI) design is not ad hoc, but a deliberate and well-justified component of our framework. While the form of injection differs from that in stochastic DDIM, the core motivation remains consistent, introducing controlled stochasticity to enhance generative diversity. We argue that applying noise to attention components is both meaningful and effective, especially for non-rigid editing tasks, and we will elaborate on the reasoning and benefits of this design below.
>
> 1. **Motivation: Enhancing Generative Diversity via Controlled Stochasticity.**
> Our core motivation is to introduce controlled, targeted stochasticity into the self-attention mechanism to enhance generative diversity, which is especially critical for complex non-rigid editing tasks. Unlike methods that inject noise into the entire latent variable, perturbing the Query (Q) and Key (K) vectors allows the model to explore diverse local semantic associations without disrupting the overall image structure. This provides the necessary flexibility for challenging edits precisely where it is most impactful.
> 2. **Formulation.**
> Our method perturbs the attention mechanism by injecting zero-mean Gaussian noise into the Q and K, i.e., $\mathbf{Q}' = \mathbf{Q} + \sigma_q \cdot \epsilon_q$ and $\mathbf{K}' = \mathbf{K} + \sigma_k \cdot \epsilon_k$, where $\epsilon_q, \epsilon_k \sim \mathcal{N}(0, \mathbf{I})$ and $\sigma_q, \sigma_k$ are scalar coefficients that control the noise strength. The attention is then computed as $\text{softmax}\left(\frac{\mathbf{Q}'{\mathbf{K}'}^\top}{\sqrt{d}}\right)\mathbf{V}$.
> We deliberately choose to inject noise before the softmax rather than after, for two reasons: First, for **stability and controllability**: by injecting noise pre-softmax, any perturbations are normalized, leading to only smooth, localized changes in the attention distribution. This ensures the effect is bounded and stable, avoiding the severe content distortions that could arise from adding noise directly to the Value (V) vectors. Second, this approach provides **semantic guidance** rather than content distortion. Perturbing Q and K affects where the model attends, not what content it attends to. This enables a more targeted exploration of token relationships to better align with the target prompt, a crucial capability for edits requiring fine-grained semantic adjustments.
> 3. **Empirical Validation.**
> Our empirical results strongly validate this design's effectiveness. SNI significantly improves the model's semantic alignment with the target prompt, particularly for non-rigid transformations of pose or shape, while maintaining high consistency in the background and unedited regions. This confirms that our stochastic attention serves as a lightweight yet potent inductive bias, providing essential flexibility for challenging generative tasks.
>
> **[W4: Paper Structure Suggestion]**
> Thank you for the suggestion. We appreciate your positive feedback on the supplementary material, and will incorporate key parts in the main paper.
>
> References
>
> [1] Xuan Ju, et al. Direct inversion: Boosting diffusion-based editing with 3 lines of code. in ICLR (2024)
>
> [2] Mokady, et al. Null-text inversion for editing real images using guided diffusion models. in CVPR (2023).
>
> [3] Huberman-Spiegelglas, et al. An edit friendly ddpm noise space: Inversion and manipulations. in CVPR (2024).

---

> ### Comment · Reviewer_9JHL · 2025-08-01
> **Thanks for the rebuttal**
>
> I appreciate the authors’ effort during the rebuttal process. My concerns are now partially resolved, especially regarding the following points:
>
> **W1.1.** I now understand that for non-rigid editing tasks, commonly used metrics (*e.g.* Distance, PSNR, LPIPS, etc …) do not fully represent editing performance. As the authors mentioned, the 1) Under-Editing Trap and 2) Metric Scaling Trap may lead to some misleading evaluations, making the task evaluation challenging–which is an issue well  illustrated in Table 1 of the rebuttal. I also reviewed the figures in the main paper pointed out by the authors, including Figure 6, and they helped clarify the effectiveness of the method.
>
> **W1.2.** Thank you for the quantitative evaluation on computational overhead. I agree with the authors’ claim that FSI-Edit achieves a favorable tradeoff between its performance and computational efficiency. It’s noticeable that each component contributes meaningfully to overall performance without inducing significant additional runtime.
>
> **W2.** Thanks for the direct comparison across diverse reconstruction-refining methods. As demonstrated in Table 3 and 4 of the rebuttal, ITN achieves on-par or superior performance compared to various baselines (DDIM, Direct inversion, Null-text inversion, and Edit-friendly inversion), which supports its significance and effectiveness.
>
> **W3.** Thanks for clarification regarding the motivation and formulation of SNI. I find the idea of injecting noise into Q and K particularly novel and interesting. The authors’ justification for excluding the V map for noise injection, as well as the reason for injecting noise before softmax rather than after–focused on stability, controllability, and semantic guidnace–resolved my previous questions. Empirical validation, including Figure 6 of the main paper, further strengthens the effectiveness of the proposed component.
>
> However, I still have one remaining question:
>
> **W1.1.** (Follow-up) Could authors report ablation study results for **rigid-editing tasks** – such as those shown in the right side of Figure 1 of the main paper? I think this would help fully address my concern regarding the ablation study by demonstrating the necessity and contribution of each component in rigid-editing scenarios.

---

> > ### Author Response · Authors · 2025-08-02
> >
> > Thank you for this important follow-up question. We appreciate the opportunity to provide additional evidence of our framework’s effectiveness under rigid editing scenarios. The results are summarized in the table below.
> >
> > Table 1: Ablation Study on Rigid Tasks
> > |Method|Distance×10³↓|PSNR↑|LPIPS×10³↓|MSE×10⁴↓|SSIM×10²↑|CLIP Whole↑|CLIP Edited↑|
> > |-|-|-|-|-|-|-|-|
> > |Baseline|11.68|28.35|70.98|24.02|88.21|24.40|22.08|
> > |FRF|11.11|28.35|69.78|24.17|88.70|25.11|22.46|
> > |SNI|13.71|27.23|79.42|30.77|87.65|25.61|22.84|
> > |FRF+SNI|11.71|27.87|71.08|26.62|87.99|25.28|22.61|
> > |FRF+SNI+ITN|11.55|27.91|71.12|26.45|88.16|25.37|22.68|
> >
> > **FRF** simultaneously improves background preservation and semantic alignment.
> > **SNI** increases editing flexibility, but introduces some degradation in pixel-level metrics.
> > **FRF+SNI** combined provides a balance to rigid editing tasks.
> > **FRF+SNI+ITN** further refines the results, demonstrating ITN’s role in integrating edits while preserving structural coherence.
> >
> > The results show that each module contributes distinct advantages under rigid editing tasks. FRF enhances structural preservation, SNI injects necessary variability, and ITN refines the output through better inversion. Together, these modules enable robust and flexible editing capabilities. We sincerely hope this resolves your concern and appreciate your constructive feedback.

---

> > > ### Comment · Reviewer_9JHL · 2025-08-03
> > > **Thanks for the response**
> > >
> > > Dear authors,
> > >
> > > Thank you for providing the additional ablation study under rigid editing scenarios. My concern has been well addressed. As shown in Table 1 and explained by the authors, each component plays an important role in image editing , and using all components together gives the best CLIP similarity. This indicates that the edited images are semantically well-aligned with the target prompt. I would appreciate it if the authors could further discuss ablation study with qualitative examples in the final version of the manuscript. In conclusion, I have increased my rating to 4.

---

> > > > ### Author Response · Authors · 2025-08-03
> > > >
> > > > Thank you very much for your positive feedback. We greatly appreciate your valuable suggestions and will make sure to include qualitative examples of the ablation study in the final version of the manuscript.

---

### Note · Authors · 2025-08-12

We sincerely thank all reviewers, the AC, and the SAC for their valuable time, constructive feedback, and efforts throughout the review process.
Reviewers acknowledged the novelty of frequency-domain features and novel components (#9JHL), the well-motivated integration of frequency and stochasticity (#Xe12), and the unified framework enabling faithful rigid and non-rigid editing (#7PE5).
After the rebuttal, all three reviewers who initially had concerns acknowledged our clarifications and raised their ratings:

**9JHL (3 → 4)**

>**Concern:** (1) Interpretation of ablation study results; (2) ITN module lacked direct comparison.

>**Response:** For (1), we explained the difference between non-rigid editing metrics and actual visual performance, and supplemented rigid-editing ablation results. For (2), we provided direct comparisons of ITN against reconstruction-refining methods. The reviewer accepted our clarifications and raised the score.

**Xe12 (Score Raised)**

>**Concern:** (1) Gap to SOTA; (2) Interpretation of ablation study results.

>**Response:** For (1), we added an average-rank comparison across all metrics, showing our method achieves overall SOTA performance, and conducted a user study demonstrating superiority. For (2), we explained the metric–performance gap in non-rigid editing and clarified how ablation results should be interpreted in this context. The reviewer acknowledged our explanations and indicated an increased score.

**7PE5 (4 → 5)**

>**Concern:** Mixed quantitative results.

>**Response:** We conducted a user study as suggested, confirming the advantage of our method. The reviewer accepted the results and raised the score.

**MHta (4)**
>The reviewer indicated not being deeply familiar with this research area and therefore did not raise specific concerns.

---

### Decision · Program_Chairs · 2025-09-17

**Decision:**

Accept (poster)

**Comment:**

This paper presents FSI-Edit, a novel framework for flexible image editing. It uses frequency-domain fusion to preserve source details while allowing significant structural changes. It also injects noise to increase generative diversity. The method is technically sound, and it tackles the challenging problem of non-rigid editing. The experiments show strong performance across different model architectures. While initial reviews noted weaknesses in the ablation studies and mixed quantitative results, the paper's novelty and impressive qualitative results are significant.

During the discussion period, reviewers raised several key concerns. For example, Reviewers 9JHL and Xe12 questioned the necessity of each proposed component. Reviewers Xe12 and 7PE5 noted that quantitative results were mixed. Reviewer 9JHL also requested better justification for the noise injection module and more comparisons. The authors provided an informative rebuttal, clarifying that standard metrics can be misleading for non-rigid edits. They also conducted a new user study, which strongly favored their method. The authors provided additional ablation results for rigid edits that clearly showed the contribution of each component. These comprehensive responses successfully addressed all major concerns, leading multiple reviewers to raise their scores. I agree with the reviews and recommend accepting this paper.